

# Seismicity and seismotectonics of the Albstadt Shear Zone in the northern Alpine foreland

Sarah Mader[1,*], Joachim R. R. Ritter[1], Klaus Reicherter[2] and the AlpArray Working Group[3]

5  [1] Karlsruhe Institute of Technology, Geophysical Institute, Hertzstr. 16, 76187 Karlsruhe, Germany
[2] RWTH Aachen University, Institute of Neotectonics and Natural Hazards Group, Lochnerstr. 4 – 20, 52056 Aachen, Germany
[3] http://www.alparray.ethz.ch

[*] corresponding author: sarah.mader@kit.edu





**Abstract.** The region around the town Albstadt, SW Germany, was struck by four damaging earthquakes with magnitudes greater than five during the last century. Those earthquakes occurred along the Albstadt Shear Zone (ASZ) which is characterized by more or less continuous microseismicity. As there are no surface ruptures visible which may be connected to the fault zone, its characteristics can only be studied by its seismicity. We use the earthquake data of the state earthquake service of Baden-Württemberg from 2011 to 2018 and complement it with additional phase picks beginning 2016 at the AlpArray and StressTransfer seismic networks in the vicinity of the ASZ. This extended dataset is used to determine new minimum 1-D seismic *vp* and *vs* velocity models and corresponding station delay times for earthquake relocation. Fault plane solutions are determined for selected events and the direction of the maximum horizontal stress is derived.

The minimum 1-D seismic velocity models have a simple and stable layering with increasing velocity with depth in the upper crust. The corresponding station delay times can be well explained by the lateral depth variation of the crystalline basement. The relocated events align north-south with most of the seismic activity between the towns of Tübingen and Albstadt east of the 9° E meridian. The events can be separated into several subclusters which indicate a segmentation of the ASZ. The majority of the 36 determined fault plane solutions features a NNE-SSW strike, but also NNW-SSE striking fault planes are observed. The main fault plane associated with the ASZ is dipping steeply and the rake indicates mainly sinistral strike-slip, but we also find minor components of normal and reverse faulting. The determined direction of the maximum horizontal stress of 147° is in good agreement with prior studies. This result indicates that the stress field in the area of the ASZ is mainly generated by the regional plate driving forces as well as the Alpine topography.

## 1 Introduction

The Swabian Alb near the town of Albstadt is one of the most seismically active regions in Central Europe (Grünthal and the GSHAP Region 3 Working Group, 1999). In the last century four earthquakes with magnitudes greater than 5 occurred in the region of the Albstadt Shear Zone (ASZ, Fig 1, e.g. Stange and Brüstle, 2005, Leydecker, 2011). Today such events could cause major damages with economic costs of several hundred



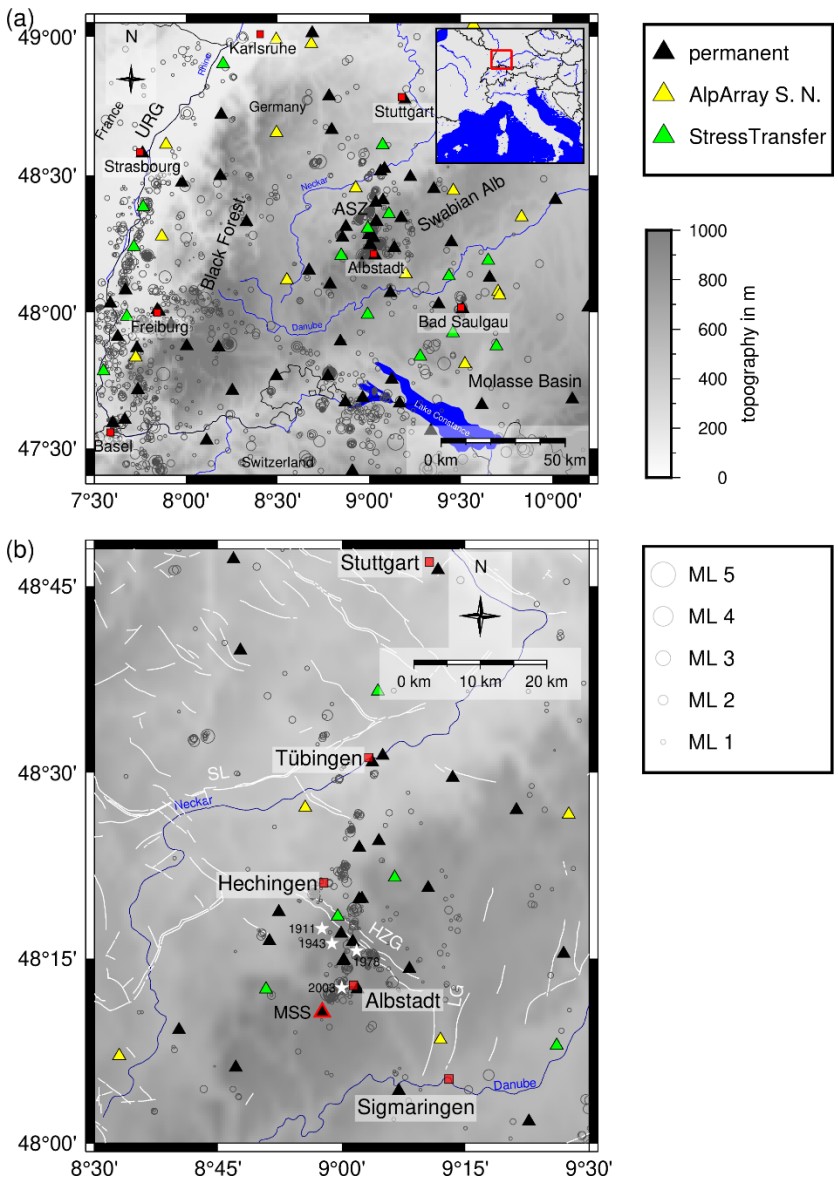

**Fig. 1 a) Overview over our research area located in the southwest of Germany in the northern Alpine foreland. The ASZ is our research target. Black triangles represent permanent seismic stations of the LED and other agencies. Yellow triangles represent temporary AlpArray seismic stations. Green triangles display the fifteen temporary seismic stations of the StressTransfer network. The gray circles display the seismicity scaled by magnitude from 2011 to 2018. URG=Upper Rhine Graben**

**b) Close up of the area of the ASZ. Symbols like in Fig. 1a. Red framed triangle highlights central station Meßstetten (MSS) of the minimum 1-D seismic velocity model. White stars mark epicenters of the 4 strongest events with a magnitude greater than 5 in 1911, two in 1943 (same epicenter) and 1978, as well as the earthquake on 22nd March 2003 with a local magnitude of 4.4 (Leydecker, 2011). White lines indicate known and assumed faults (Regierungspräsidium Freiburg, Landesamt für Geologie, Rohstoffe und Bergbau (Hrsg.), 2019). The Hohenzollern Graben (HZG) is the only clear morphological feature in the close vicinity to the ASZ. Other large tectonic features are the Lauchertgraben (LG) and the Swabian Line (SL).**

**Topography is based on the ETOPO1 Global Relief Model (Amante and Eakins, 2009).**





million Euros (Tyagunov et al., 2006). Although the earthquakes caused major damage, no surface ruptures have been found
        or described (e.g. Schneider, 1971). For this reason, the ASZ can only be analyzed by its seismicity to derive its geometry,
        possible segmentation, and faulting pattern. One of the best observed earthquakes happened on 22nd March 2003 and it was
        described as a sinistral strike-slip fault with a strike of 16° (Stange and Brüstle, 2005), which is similar to the observations
        from prior events (e.g. Schneider, 1973, Turnovsky, 1981, Kunze, 1982). Since 2005 the seismic station network of the state

earthquake service of Baden-Württemberg (LED) was changed and extended (Stange, 2018) and in summer 2015 the
        installation of the Alp Array Seismic Network (AASN) started (Hetényi et al., 2018). In 2018 we started our project
        StressTransfer, in which we investigate areas of distinct seismicity in the northern Alpine foreland of Southwest Germany and
        the related stress field. For this study we installed additional 15 seismic stations, equipped with instruments of the KArlsruhe
        BroadBand Array (KABBA), in our research area (Fig. 1a).

Here we present a compilation of different datasets to refine hypocentral parameters of the ASZ. For this we analyze the
        earthquake catalog of the LED from 2011 to 2018 (Bulletin-Files des Landeserdbebendienstes B-W, 2018) and complement it
        with additional phase picks from recordings of AASN (AlpArray Seismic Network, 2015) and our own StressTransfer seismic
        stations. We calculate a new minimum 1-D seismic velocity model and relocate the events. For some relocated events we
        calculate fault plane solutions. This procedure gives us a new view of the geometry of the ASZ based on its permanent

microseismic activity. Furthermore, we use the fault plane solutions to derive the orientation of the maximum horizontal stress
        $S_{Hmax}$ in the area of the ASZ and discuss it with known results.

## 2 Geological and Tectonic Setting

Southwest Germany is an area of low to moderate seismicity. The most active fault zones are the Upper Rhine Graben (URG)
and the area of the ASZ and the Hohenzollern Graben system (HZG, Fig. 1b). In the region of the URG the seismicity is

distributed over a large area. In comparison, in our research area the seismicity clusters in the close area around the ASZ and
        the HZG.
        The ASZ is named after the town of Albstadt situated on the Swabian Alb, a mountain range in South Germany (Fig. 1a).
        South Germany consists of several tectono-stratigraphic units, a polymetamorphic basement with a Mesozoic cover tilted
        towards southeast to east due to extension in the URG (Fig. 1a), associated with updoming (Reicherter et al., 2008, Meschede

and Warr, 2019). The URG forms the western tectonic boundary, whereas the eastern boundary comprises the crystalline
        basement of the Bohemian Massif. To the south the foreland basin of the Alps (Molasse Basin, Fig. 1a) frames the area in a
        triangular shape. The Molasse Basin covers the whole area south of the Swabian Alb up to the Alpine mountain chain. It is
        filled with terrestrial, freshwater and shallow marine sediments (Meschede and Warr, 2019). The Swabian Alb is bounded by
        the rivers Neckar in the north and Danube in the south (Fig. 1a). The sedimentary layers of the Swabian Alb, which consist of

Jurassic limestone, marl, silt and clay, dip downwards by 4° to the southeast and disappear below the Molasse Basin (Fig. 1a)
        and the Alpine Mountain Chain (Meschede and Warr, 2019). The sedimentary cover of the Swabian Alb forms a typical cuesta





landscape with major escarpments built up by resistant carbonates of the Late Jurassic that is cut by several large fault systems, which are detectable in the present-day topography (Reicherter et al., 2008). The Black Forest to the west of the Swabian Alb experienced the most extensive uplift due to the extension of the URG. Here, even metamorphic and magmatic rocks of the

Paleozoic basement are exposed. To the north and northwest of the Swabian Alb, Triassic rocks are cropping out (Meschede and Warr, 2019). Due to the different uplift and erosional states of South Germany the depth of the crystalline basement varies strongly between -5.4 km and 1.2 km above sea level (Rupf and Nitsch, 2008).

The regional stress field of southwest Germany is dominated by an average horizontal stress orientation of 150° (e.g. Müller et al., 1992, Reinecker et al., 2010, Heidbach et al., 2016) and was determined from focal mechanism solutions, overcoring,

borehole breakouts and hydraulic fracturing (e.g. Kastrup et al., 2004, Reiter et al., 2015, Heidbach et al., 2016). It is characterized by NW-SE horizontal compression and NE-SW extension (e.g. Kastrup et al., 2004) and developed during late Miocene (Becker 1993). Analysis of several kinematic indicators hint that fault planes where already activated repeatedly during the Cenozoic (Reicherter et al., 2008). Three main groups of fault planes can be observed. First, NNE-SSW to N-S striking mainly sinistral fault planes, which are similar to the ASZ or the Lauchertgraben (Fig. 1b) and parallel the URG.

Second, NW-SE striking normal and/or dextral strike-slip fault planes, which correspond to the HZG in our area. Older kinematic indicators, like fibre tension gashes and stylolites, hint a sinistral initiation of those NW-SE striking fault planes during the Late Cretaceous-Paleogene with a maximum horizontal compression in NE-SW direction (Reicherter et al., 2008). And third, ENE-WSW oriented fault planes, which are mainly inactive, but some exhibit dextral strike-slip or reverse movement, for example the Swabian Line (Schwäbisches Lineament, Fig. 1b). The stress field in our research area is quite

constant, except of close to the southern part and within the HZG. There the $S_{Hmax}$ direction rotates about 20° counterclockwise into the strike of the HZG (130°, Baumann, 1986), which may be caused by a reduced marginal shear resistance.

The only morphologically visible tectonic feature close to Albstadt is the HZG (Fig. 1b), a small graben with an inversion of relief and a NW-SE strike (Schädel, 1976, Reinecker and Schneider, 2002). The 25 km long HZG has dip angles between 60-70° at the main faults and a maximum graben width of 1.5 km, which leads to a convergence depth of the main faults in 2-3

km depth (Schädel, 1976). Thus, the HZG is interpreted as a rather shallow tectonic feature. To the north and south of Albstadt there are further similar graben structures like the HZG, namely the Fildergraben, Rottenburg Flexur, western Lake Constanz, Hegau which are also about parallel to the main horizontal stress field (Reinecker et al., 2010) like the HZG. Reinecker and Schneider (2002) propose a tectonic model to relate the graben structures with the ASZ below. In their model, the graben structures are en-echelon features compensating the displacement of the ASZ. The ASZ itself is partly decoupled from the

surface by a layer of evaporites of the salinar formation of the Middle Triassic from the overlying sedimentary layers (Reinecker and Schneider, 2002). Stange and Brüstle (2005) consider the bottom of the Mesozoic sediments as mechanical decoupling horizon as no earthquakes occur above 2 km depth.

Another tectonic feature in our research area is the ENE-WSW striking Swabian Line north of the river Neckar. It extends from the Black Forest area partly parallel along the cuesta of the Swabian Alb to the east (Reicherter et al., 2008, Fig. 1b). The



sense of movement along the Swabian Line is dextral. To the east of the ASZ near Sigmaringen the Lauchertgraben strikes
north-south, around parallel to the ASZ with a sinistral sense of displacement (Geyer and Gwinner, 2011, Fig. 1b).

The faults in southwest Germany exhibit mainly moderate displacements. At the HZG, for example, the maximum vertical
offset is of the order of 100 to 150 m. The horizontal offset is considerably lower and more difficult to determine (Reicherter
et al., 2008).

Wetzel and Franzke (2003) identified a 5 to 10 km broad zone along the 9° E meridian of lineations striking predominantly N-
S, NW–SW and ENE-WSE pursuable from Stuttgart to Lake Constance. The graben structures, like the HZG, are limit by the
NW-SE striking faults (Reicherter et al., 2008). Those faults are expected to be possibly active at intersections with N-S
striking faults due to a reduction in shear resistance accompanied by aseismic creep (Schneider, 1979, Schneider, 1993, Wetzel
and Franzke, 2003).

The first documented earthquakes in the area of the ASZ occurred in 1655 near Tübingen and had an intensity of 7 to 7.5
(Leydecker, 2011). A similarly strong earthquake occurred in 1911 near Albstadt-Ebingen with a local magnitude of 6.1 (Fig.
1b, Leydecker, 2011), causing damage buildings (Reicherter et al., 2008). The seismic shock triggered landslides with surface
scarps in both the superficial Quaternary deposits as well as the Tertiary Molasse sediments (Sieberg and Lais, 1925) in the
epicentral area and close to Lake Constance, demonstrating the potential of hazardous secondary earthquake effects (Reicherter

et al., 2008). Since the 1911 earthquake the Swabian Alb is one of the most seismically active regions in the northern Alpine
foreland with in total three more earthquakes with a local magnitude greater 5 (Fig. 1b, 2x 1943, 1978, e.g. Reinecker and
Schneider, 2002, Stange and Brüstle, 2005). The latest strong events occurred on 4th November 2019 and 27th January 2020,
with ML of 3.8 and 3.5 (Regierungspräsidium Freiburg, Landesamt für Geologie, Rohstoffe und Bergbau, 2020). The average
seismic dislocation rates along the ASZ are on the order of 0.1 mm/a (Schneider, 1993). The return period of earthquakes along

the ASZ with a magnitude of 5 is approximately 1000 years (Schneider, 1980). From aftershock analyses and focal mechanism
calculations we know that the ASZ is a steep NNE to SSW oriented sinistral strike-slip fault (e.g. Haessler et al., 1980,
Turnovsky, 1981, Stange and Brüstle, 2005) in the crystalline basement, as all earthquakes occur in a depth greater than 2 km
(Stange and Brüstle, 2005). The N-S extension of the fault zone is still under debate: Reinecker and Schneider (2002) propose
an extension from Northern Switzerland towards north to the area of Stuttgart, whereas Stange and Brüstle (2005) do not find

this large extension as most of the seismicity happens on the Swabian Alb.

## 3 Earthquake Data & Station Network

As basis for our study we use the earthquake catalog of the LED from 2011 to 2018 for earthquakes within the area close to
the ASZ (8.5-9.5° E, 48-48.8° N, Fig. 1b). For these 575 earthquakes we received the bulletin files of the LED (Bulletin-Files
des Landeserdbebendienstes B-W, 2018), consisting of hypocenter location, origin time, local magnitude ML and all phase

travel time picks with corresponding quality and P-phase polarity. The LED picks from 2011 to 2018 are from 51 LED seismic
stations and 14 seismic stations run by other agencies like the state earthquake service of Switzerland (Fig. 1a).



Additionally, within the AlpArray Project (Hetényi et al., 2018), nine seismic stations were installed beginning in summer 2015 within 80 km distance to the ASZ, four of them directly around the ASZ (AlpArray Seismic Network, 2015, Fig. 1b). To get an even denser network and to detect microseismicity we started to install another 15 seismic broadband stations from the

KABBA beginning in July 2018 in areas with striking seismicity in the northern Alpine foreland within our project StressTransfer (Fig. 1a). Five of those stations are located in the vicinity of the ASZ (Fig. 1b) and three of them where already running at the end of 2018.

We complemented the LED catalog with additional seismic P- and S-phase picks from the four AASN stations located around the ASZ from 2016 to 2018 and our StressTransfer stations recording in 2018. In total, our combined dataset consists of 575

earthquakes (Fig. 1b) with 4521 direct P- and 4567 direct S-phase travel time picks from 69 seismic stations.

## 4 Data Processing

### 4.1 Phase Picking

To complement the LED catalog, we use a self-written code in ObsPy (e.g. Beyreuther et al., 2010) for semi-automatic manual picking of the direct P- and S-phases. The raw waveform recordings are bandpass filtered with a zero-phase four corners

butterworth filter from 3 to 15 Hz. Using the hypocenter coordinates of the LED we calculate an approximate arrival time at a seismic station. Around this arrival time, we define a noise and a signal time window following Diehl et al. (2012), so that we can calculate the signal to noise ratio (SNR) of our phase onsets. Our code calculates automatically the earliest possible pick (ep) and the latest possible pick (lp) (Diehl et al., 2012) to get consistent error boundaries for each pick. Finally, the error boundaries are checked, and the phase pick is done manually between the two error boundaries. The qualities of 0 up to 4 of

the picked arrival time are set depending on the time difference between ep and lp (Table A1). For consistency a similar relationship is used between picking quality and uncertainty as defined by the LED (Bulletin-Files des Landeserdbebendienstes B-W, 2018).

### 4.2 Inversion for Minimum 1-D Seismic Velocity Models with VELEST

The LED uses the program HYPOPLUS (Stange and Brüstle, 2005) for routine location, with which one can apply a 1.5-

dimensional approach by using several 1-D seismic velocity models for selected regions (Stange and Brüstle, 2005, Bulletin-Files des Landeserdbebendienstes B-W, 2018). They use two P-wave velocity (*vp*) models, a Swabian Jura model and a model for the state of Baden-Württemberg, and they define the S-wave velocity (*vs*) model by a constant *vp/vs*-ratio (Stange and Brüstle, 2005, Bulletin-Files des Landeserdbebendienstes B-W, 2018, Fig. 4a) & b)). Furthermore, no station delay times are used (Bulletin-Files des Landeserdbebendienstes B-W, 2018).

To determine a complemented catalog, we invert for new minimum 1-D seismic *vp* and *vs* models in the region of the ASZ with station delay times, using the program VELEST (Kissling et al., 1994, Kissling et al., 1995, VELEST Version 4.5). As central recording station we chose the station Meßstetten (MSS, Fig. 1b) as this station was running during our complete





observational period and it is the oldest seismic recording site on the Swabian Alb, recording since 2$^{nd}$ June 1933 (Hiller, 1933). To get the best subset of our catalog for the inversion, we select only events with at least eight P-arrival times for the

inversion for *vp* and either eight P- or eight S- arrival times for the simultaneous inversion for *vp* and *vs*. The P-pick times exhibit a quality of 1 or better and the S-picks need a quality of 2 or better (Table A1). Events outside of the region of interest, 48.17 °N – 48.50 °N and 8.75 °E – 9.15 °E, with an azimuthal GAP greater than 150° and an epicentral distance of more than 80 km are rejected. This selection leads to a high-quality subset of 68 events with 789 P-phase picks for *vp* and 99 events with 945 P-picks and 1019 S-phase picks for the *vp* and *vs* inversion (Fig. 2). We apply a staggered inversion scheme after Kissling

et al. (1995) and Gräber (1993), first inverting for *vp* and then for *vp* and *vs* together while damping the *vp* model. The inversion for station delay times was done after the change in velocity was stable.

**Fig. 2 Ray coverage and input data set for the inversion with VELEST. White circles represent the 99 selected events which are used for *vp* and *vs* inversion. Seismic stations are indicated as triangles and color-coded with the number of picks at a station. Topography**
**is based on the ETOPO1 Global Relief Model (Amante and Eakins, 2009).**




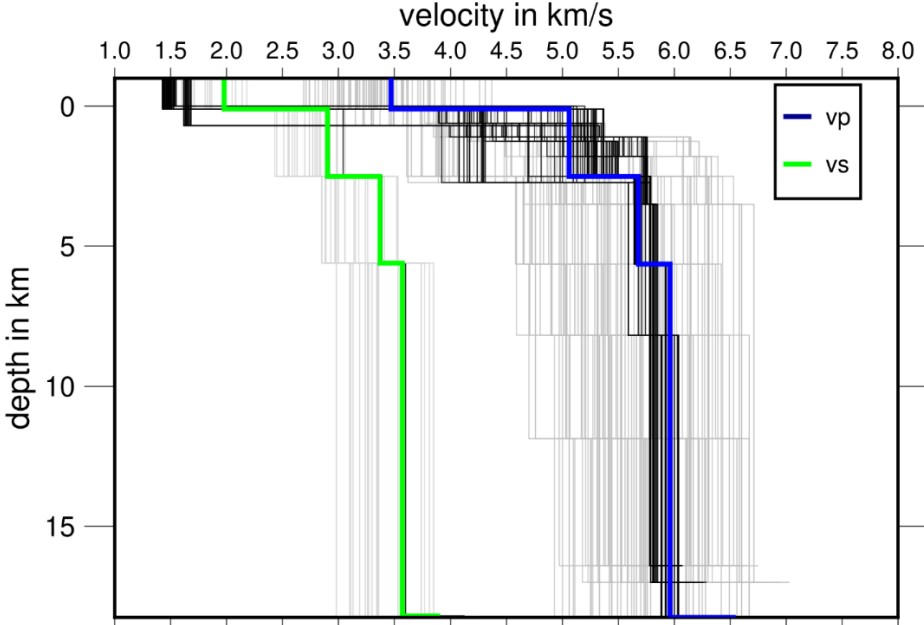

**Fig. 3 VELEST input models for *vp* (84) and *vs* (21) (gray) and output *vp* (84) and *vs* (21) after inversion (black) together with the chosen ASZmod1 (colored). A good convergence of the models can be observed, especially for *vs*. The second layer converges worst. An instability of the first layer with a tendency to unrealistic low seismic velocities can be seen. For this reason, the velocity of ASZmod1 was fixed in the first layer.**

To probe our seismic velocity model space, inversions with 84 different starting models are calculated with 4 differently layered models from seismic refraction profile interpretations (Gajewski and Prodehl, 1985, Gajewski et al., 1987, Aichroth et al., 1992), the LED Swabian Jura model (Stange and Brüstle, 2005) and realistic random *vp* variations (Fig. 3). The minimum 1-D *vp* model with the smallest RMS and the simplest layering was selected as final *vp* model for the simultaneous *vp* and *vs* inversion. Together with a *vp/vs*–ratio of 1.69 (Stange and Brüstle, 2005) it was also used to calculate the *vs* starting model, which was randomly changed to get in total 21 *vs* starting models (Fig. 3). The resulting minimum 1-D *vp* and *vs* models (ASZmod1, Fig. 4) were selected due to their small RMS.

To test the stability of ASZmod1 we randomly shifted all 99 events in space by maximum 0.1° horizontally and 5 km with depth (Kissling et al., 1995). The result of this shifttest demonstrates that we can determine stable hypocenters, with an average deviation of less than 0.005° horizontally and of less than 2 km in depth for more than 90% of the events in the catalog (Fig. S1). The seismic velocities are stable except for the first and second layer (Fig. S2a and b). The first layer was instable already during the inversion process (Fig. 3), therefore we damped its layer velocities and set them to realistic *vp* and *vs* values, based on the seismic *vp* of the refraction profile interpretations (Gajewski et al., 1987). The instability in both upper layers may be caused by few refracting rays and therefore small horizontal raylengths through the layers as well as only few earthquakes





within these layers (Fig. S2c). In total, the stability test (Fig. S1 and S2) indicates that the model represents the data and region very well and that the determined hypocenter locations are stable.

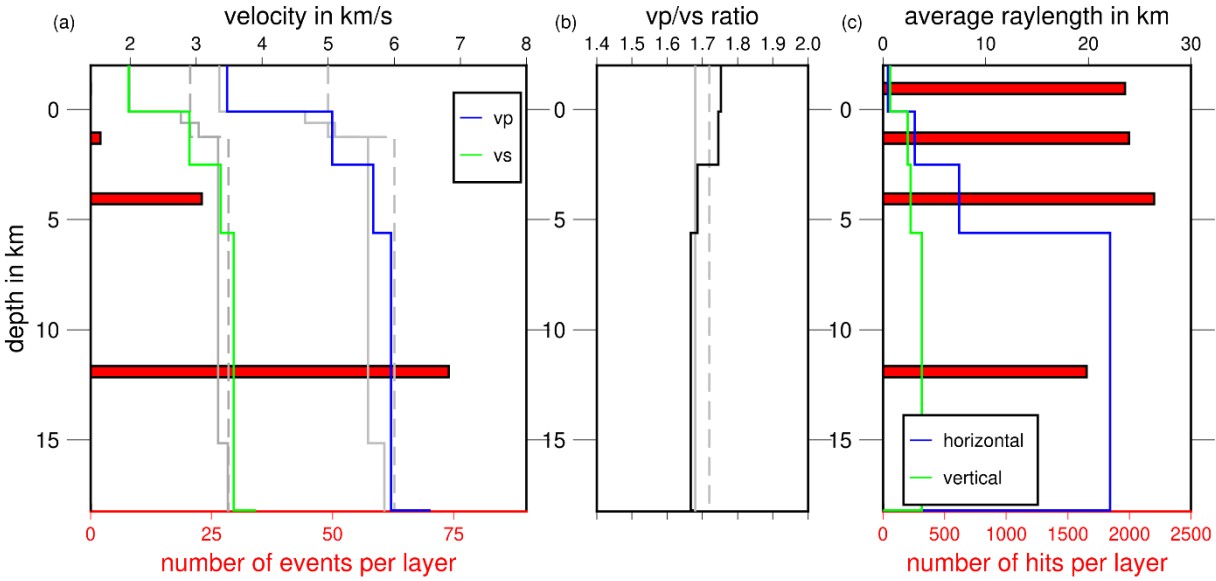


**Fig. 4 a) Final minimum 1-D seismic velocity models (ASZmod1), *vs* in green, *vp* in blue. Gray lines represent velocity models of the LED (Bulletin-Files des Landeserdbebendienstes B-W), solid: Swabian Jura models, dashed: Baden-Württemberg models. Red bars are scaled with the number of events in each layer of the velocity model.**

**b) *vp/vs*-ratio of ASZmod2 and the LED models.**

**c) Ray statistics of used ray paths. Red bars display number of hits per layer. Blue and green line give the average horizontal and vertical raylength.**

We calculated an error estimate based on the variation of the 21 output *vp* and *vs* models with our chosen layer model of

Gajewski et al. (1987) for ASZmod1 (Table 1). For a precise estimation we determined two times the standard deviation (2σ) of the velocity models for each layer. For the uppermost layer we could not estimate any error, as the first layer was manually set and strongly damped during inversion process. The 2σ range is small for the third and fourth layer. This was expected as most of the events are located within those layers and as all other models, also with different layering, converge to similar velocities in those layers (Fig. 3). The error estimate for the second layer has to be considered carefully as this layer revealed

strong instabilities during the stability test (Fig. S2). The fifth layer also has larger 2σ uncertainties relative to layers three and four, which is caused by less ray coverage and no events located below 18.25 km depth.





|         | Layer top in km | *vp* in km/s | 2σ *vp* in km/s | *vs* in km/s | 2σ *vs* in km/s |
|---------|-----------------|--------------|-----------------|--------------|-----------------|
| Layer 1 | -2              | 3.47         | -               | 1.98         | -               |
| Layer 2 | 0.1             | 5.06         | 0.30            | 2.90         | 0.06            |
| Layer 3 | 2.5             | 5.68         | 0.03            | 3.37         | 0.01            |
| Layer 4 | 5.63            | 5.95         | 0.02            | 3.57         | 0.01            |
| Layer 5 | 18.25           | 6.55         | 0.31            | 3.91         | 0.32            |

**Table 1 ASZmod1 with corresponding error estimates based on 2σ.**

### 4.3 Relocation, Station Corrections and Error Estimation with NonLinLoc

To relocalize the complete earthquake catalog we use the program NonLinLoc (NLL, Lomax et al., 2000), a non-linear oct-
tree search algorithm. NLL calculates travel time tables after the eikonal finite-difference scheme of Podvin and Lecomte
(1991) on a predefined grid, here with 1 km grid spacing. With the implemented oct-tree search algorithm (density) plots of
the probability density function (pdf) of each event are determined after the inversion approach of Tarantola and Valette (1982)
with either the L2-RMS likelihood function (L2) or the Equal Differential Time likelihood function (EDT). The determined
pdf contains location uncertainties due to phase arrival time errors, theoretical travel time estimation errors, and the geometry
of the network (Husen et al., 2003). Based on the pdfs an error ellipsoid (68% confidence) is determined, which we use to
calculate latitude, longitude, and depth error estimates for each earthquake (Fig. 5). The estimated errors of our events are
getting smaller beginning in 2016, especially the depth error estimate. This reduction correlates well with the increased number
of picks per event, so with the increased number of seismic stations around the ASZ due to the modification of the LED
network and the installation of the AASN and the StressTransfer stations (Hetényi et al., 2018, Stange, 2018, Fig. 5). As final
hypocenter solution the maximum likelihood hypocenter is selected, which corresponds to the minimum of the pdf.

We compared the resulting hypocenters and error estimates using the L2 or the EDT likelihood function. The comparison
mainly indicates similar earthquake locations, but we find EDT errors (Fig. S3) for many events which are too large and also
unrealistic (some greater than 50 km, leading to hypocenter shifts across the whole region). For this reason, we decided to use
L2 for relocating our combined catalog.





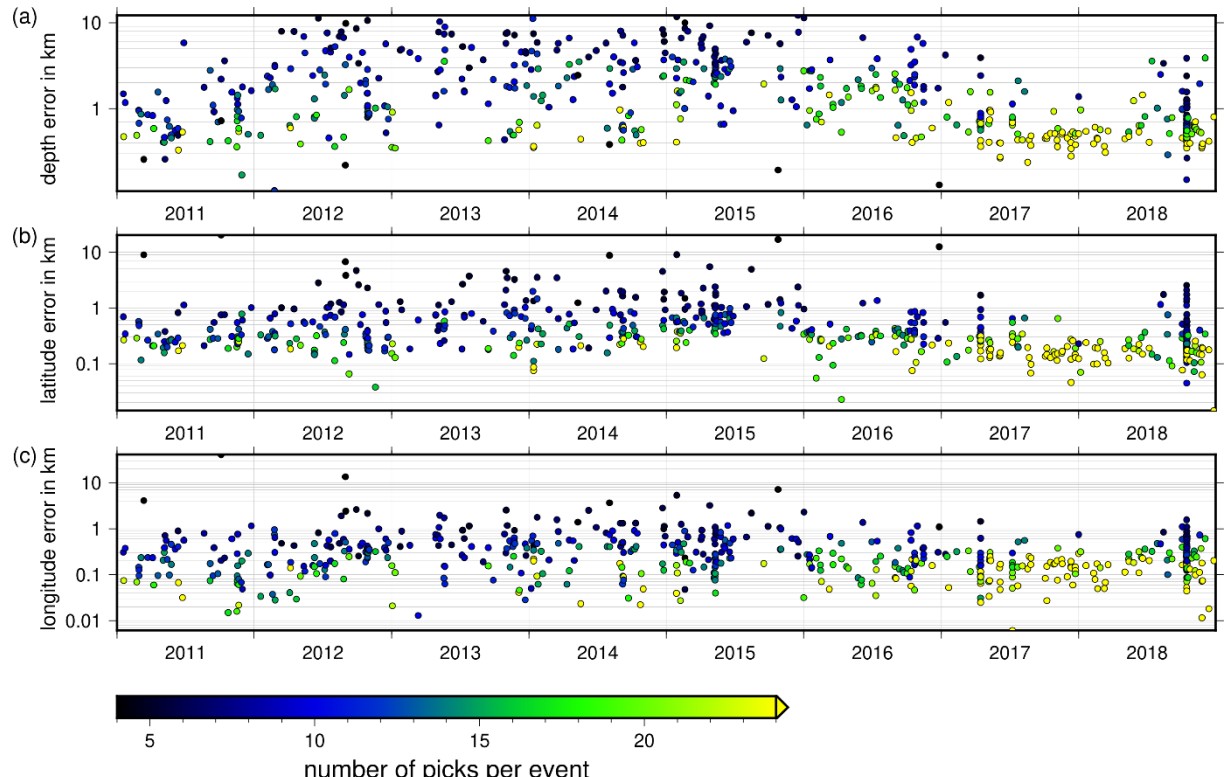

**Fig. 5 Errors calculated from the 68% confidence ellipsoid from NLL with L2 (L2-RMS likelihood function) for each event in the combined catalog for (a) depth, (b) latitude, and (c) longitude. The errors are color-coded depending on the number of picks, with dark colors indicating less picks and bright colors many picks. Hypocenters with many observations are determined with smaller errors in depth and lateral position.**

In NNL one can examine station delay times calculated from the station residuals. The station delay times are defined as the time correction subtracted from the observed P- and S- wave arrival times. This implies that negative station delay times exhibit faster velocities relative to ASZmod1 and positive station delay times exhibit slower velocities relative to ASZmod1. We used ASZmod1 and the corresponding VELEST station delay times as well as our high-quality subset of 99 earthquakes as input for NLL. After four iterative runs of NLL, with always using the output station delay times as new input station delay times, the determined station delay times become stable. As we want to relocate the whole catalog with NLL, we use for consistency the with NLL updated VELEST station delay times. Since ASZmod1 is a 1-D seismic velocity model below the reference station MSS, we expect the station delay times to become zero for MSS. After four iterative runs the actually determined station delay times of MSS are 0.014 s with σ of 0.083 s for *vp* and -0.027 s with σ of 0.064 s for *vs*. As σ is bigger than the actual station delay time and the station delay time of MSS is smaller than the maximum error range of 0.05 s of our best





**Fig 6 a) station delay times for the *vp* velocity model ASZmod1**

**b) station delay times for the *vs* velocity model ASZmod1**

Blue colored circles represent negative station delay times, indicating areas with faster velocities than ASZmod1. Red colored circles illustrate positive station delay times, indicating slower velocities than ASZmod1. Crosses are stations with zero station delays. Only stations with more than 5 travel time picks are included. Small white triangle highlights reference station MSS. Topography is based
on the ETOPO1 Global Relief Model (Amante and Eakins, 2009).





determined picks (Quality 0, Table A1), we consider the station delay times of MSS as practically zero. To account for similar small station delay times and σ, we state that all station delay times in the range of -0.05 s to 0.05 s to be practically zero station delay times, if σ is greater than the actual station delay time (Fig. 6). The fact that the NLL station delay times of MSS and surrounding stations are close to zero indicates that, even though using a different and non-linear relocation algorithm for delay

time estimation instead of VELEST, our determined minimum 1-D seismic velocity model ASZmod1 represents the seismic velocity structure below MSS and its surroundings very well.

We compared the relocated catalog with the original LED locations. Some events have large differences in hypocenter coordinates (>0.1° in latitude or longitude), which we identified as events with only a few arrival time picks (less than nine picks), a large azimuthal GAP (GAP > 180°) or wrong phase picks. Furthermore, a large deviation of expectation and maximum

likelihood hypocenters indicates an ill-conditioned inverse problem with a probable non-Gaussian distribution of the pdf (Lomax et al., 2000), which was the case for some events with only a few picked arrival times. Similar problems were also identified by Husen et al. (2003), who compared NLL locations with the routine locations of the state earthquake service of Switzerland. They also found that a good depth estimate with NLL depends on the station distances to the earthquake. Especially, for events with many observations the depth estimate was worse if the closest station was further away than the

focal depth of the event (Husen et al., 2003).

Our good located earthquakes are selected by the following criteria: more than eight travel time picks, a GAP less than 180°, a horizontal error estimate of less than 1 km, and a depth error estimate of less than 2 km (Fig. 7). Some of our good located events have quite different depth estimates compared to the LED solution (Fig. S4). Thus, we checked the station distribution for those events as proposed by Husen et al. (2003) and looked for wrong phase picks. Nevertheless, all these events have good

phase picks, a small depth error estimate, evenly distributed stations and small deviations of expectation and maximum likelihood hypocenter coordinates. For this reason, we consider our new depth locations as well determined and reliable.

In comparison with the LED catalog the majority of our relocated earthquakes are characterized by a small eastward shift as well as a stronger clustering, especially in depth (Fig. S4). The latter may result from the handset depth location for some events of the LED.





**Fig. 7** Hypocenters of the 337 best located events with a horizontal error of less than 1 km and a depth error of less than 2 km. Only events with a GAP smaller than 180° and more than eight travel time picks are included. Hypocenters are pictured as circles, color-coded with depth. All 36 focal mechanisms are displayed also color-coded with depth, red circles indicate corresponding event hypocenter. The size of the focal mechanisms is scaled depending on ML of the event. Cluster codes are placed next to the fault plane solutions. White lines indicate known and assumed faults (Regierungspräsidium Freiburg, Landesamt für Geologie, Rohstoffe und Bergbau (Hrsg.), 2019). Topography is based on the ETOPO1 Global Relief Model (Amante and Eakins, 2009).





## 4.4 Focal Mechanisms with FOCMEC

We determine fault plane solutions for 36 selected events with the program FOCMEC (Snoke, 2003), which conducts a grid
search over the complete focal sphere and outputs all possible fault plane solutions. For this we used the P-polarity picks of
the LED (Bulletin-Files des Landeserdbebendienstes B-W, 2018) and for events since 2016 we added P- and SH-polarities at
the four AASN and three StressTransfer stations. The local magnitude ML of those 36 events is in the range 0.9 to 3.4 (Table
2, Fig. 7, Bulletin-Files des Landeserdbebendienstes B-W, 2018).

Due to small magnitudes and therefore a low signal to noise ratio of the waveforms, or due to the lower station coverage before
2016, we obtain only a small number of polarity picks or an unbalanced distribution of polarities on the focal sphere. These
adversities led to many uncertain fault plane solutions. For this reason, we combined events with similar hypocenter
coordinates and polarity picks to determine a joint solution (Table 2). In this way we combined them to clusters with a
corresponding cluster code (Table 2). The cluster code hints to the location of the events and increases from south to north.
The cluster code C1 corresponds to clusters in the southern part of the ASZ close to Albstadt, whereas the cluster code C6
corresponds to events north of the river Neckar (Fig. 7). A subcode X (e.g. C1_1) subdivides clusters occurring in a specific
main cluster of the ASZ or in its vicinity. For example, C1_1 up to C1_4 are situated in the southern part of the ASZ and are
all part of the cluster C1 (Fig. 7). We also determined a fault plane solution for one isolated single event, which is assigned the
cluster code E1 (Table 2).

We compare both polarity weighting modes, relative weighting and unity weighting, of FOCMEC for all events, to explore if
the result differs significantly (Snoke, 2003). Furthermore, we checked for a dependency of the result on a single pick, by
allowing more wrong polarity picks than necessary to get a solution and compared the resulting solutions. Non-unique or
problematic cases are excluded and here we only display fault plane solutions with a good stability in both cases.

The output of FOCMEC results in a set of possible strike, dip, and rake combinations for each event. As preferred fault plane
solution the one closest to the median of strike, dip, and rake was chosen (Table 3). We use the other possible solutions to
determine uncertainties for our preferred fault plane solution. For this we recalculate all strikes to a range between 90° and
270°, to not allow large differences in strike by the transition from 360° back to 0° and by the 180° ambiguity of the strike.
We determine the 5% and 95% percentiles of strike, dip, and rake (Fig. S5) and calculate the width of the 5% to 95% percentile
range ($\Delta$strike, $\Delta$dip, $\Delta$rake, Table 3). We use the width as uncertainty range to account for a non-uniform solution distribution,
as we observe it for example for the rake of cluster C6_1 ( Fig. S5), and to assign a quality factor to the fault plane solutions
(Table A2, Table 2). We observe a low quality (3 and 4) especially for clusters consisting of only small magnitude events and
clusters with a non-uniform station distribution on the focal sphere. In Fig. 7 the fault plane solutions are displayed scaled with
magnitude and with cluster code.

| Event Origin time | ML | Cluster Code | P Picks | SH Picks | Longitude in ° | Latitude in ° | Depth in km |
|---|---|---|---|---|---|---|---|
| 110624 1028 | 2 | C3_1 | 11 | 0 | 9.04 | 48.31 | 7 |
| 131204 1941 | 2.9 | C3_1 | 13 | 0 | 9.04 | 48.31 | 6 |





| | | | | | | | |
|---|---|---|---|---|---|---|---|
| 140518 0641 | 3 | C1_4 | 16 | 0 | 8.99 | 48.20 | 5 |
| 171024 0209 | 2 | C4_2 | 11 | 0 | 9.02 | 48.35 | 9 |
| 141031 1247 | 3.2 | C1_4 | 15 | 0 | 8.99 | 48.20 | 4 |
| 150128 0004 | 3.4 | C1_4 | 19 | 0 | 8.99 | 48.20 | 5 |
| 160203 2326 | 1.1 | C5_1 | 7 | 2 | 9.05 | 48.44 | 10 |
| 160217 1925 | 1.7 | C3_1 | 11 | 2 | 9.03 | 48.29 | 4 |
| 160309 0853 | 2.3 | C1_1 | 13 | 2 | 9.01 | 48.22 | 6 |
| 160902 0757 | 2.2 | C1_1 | 14 | 3 | 9.00 | 48.21 | 5 |
| 170415 1715 | 2.1 | C4_2 | 17 | 4 | 8.96 | 48.33 | 11 |
| 170509 1618 | 2.8 | C6_1 | 22 | 3 | 8.74 | 48.55 | 14 |
| 170510 2100 | 1 | C6_1 | 10 | 3 | 8.74 | 48.55 | 14 |
| 170606 1327 | 1.8 | C1_1 | 17 | 4 | 9.00 | 48.20 | 5 |
| 170606 2058 | 1.9 | C1_1 | 25 | 4 | 9.00 | 48.20 | 5 |
| 170723 1348 | 1.4 | C5_1 | 14 | 3 | 9.04 | 48.44 | 10 |
| 170827 0559 | 1.7 | E1 | 29 | 2 | 8.87 | 48.20 | 12 |
| 171008 2254 | 1 | C2_1 | 14 | 3 | 9.03 | 48.24 | 7 |
| 171104 2311 | 1 | C6_2 | 13 | 3 | 8.98 | 48.55 | 15 |
| 171106 0039 | 1.4 | C2_1 | 20 | 3 | 9.03 | 48.24 | 7 |
| 171212 0424 | 1.4 | C1_2 | 17 | 2 | 8.99 | 48.20 | 4 |
| 171212 0539 | 2.2 | C1_2 | 22 | 3 | 8.99 | 48.20 | 4 |
| 172323 2008 | 1 | C1_3 | 15 | 3 | 8.99 | 48.20 | 3 |
| 180210 1244 | 2.3 | C4_1 | 27 | 3 | 8.95 | 48.33 | 12 |
| 180219 0829 | 1.3 | C4_1 | 17 | 3 | 8.95 | 48.33 | 12 |
| 180221 0205 | 1 | C4_1 | 13 | 3 | 8.95 | 48.33 | 12 |
| 180310 2116 | 2 | C6_2 | 18 | 2 | 8.98 | 48.55 | 15 |
| 180313 0514 | 1 | C1_3 | 11 | 3 | 8.99 | 48.20 | 3 |
| 180708 0438 | 1.1 | C2_1 | 12 | 0 | 9.03 | 48.25 | 7 |
| 181015 1501 | 1.2 | C2_2 | 18 | 3 | 9.04 | 48.25 | 10 |
| 181015 1937 | 1.3 | C2_2 | 17 | 4 | 9.04 | 48.25 | 10 |
| 181015 1941 | 1.6 | C2_2 | 27 | 4 | 9.04 | 48.25 | 10 |
| 181017 0301 | 1 | C2_2 | 14 | 4 | 9.04 | 48.25 | 10 |
| 181125 0222 | 1.4 | C2_3 | 25 | 3 | 9.04 | 48.25 | 10 |





| 181125 1026 | 0.9 | C2_3 | 13 | 5 | 9.04 | 48.25 | 10 |
| 181213 1855 | 1.1 | C4_1 | 18 | 6 | 9.02 | 48.38 | 12 |

**Table 2 Events used for focal mechanism determination with corresponding local magnitude ML, cluster code, and number of used P- and SH-phase polarities, as well as hypocenter coordinates.**


| Cluster code | strike in ° | dip in ° | rake in ° | Δstrike in ° | Δdip in ° | Δrake in ° | Strike in ° (aux) | Dip in ° (aux) | Rake in ° (aux) | quality | Stress regime | $S_{Hmax}$ orientation in ° |
|---|---|---|---|---|---|---|---|---|---|---|---|---|
| E1 | 38 | 71 | -5 | 0 | 4 | 2 | 129.63 | 85.27 | -160.93 | 0 | SS | 175 |
| C1_1 | 10 | 86 | 50 | 3 | 5 | 13 | 275.25 | 40.17 | 173.79 | 1 | U | 132 |
| C1_2 | 196 | 72 | 9 | 7 | 38 | 37 | 103.20 | 81.44 | 161.79 | 3 | SS | 148 |
| C1_3 | 166 | 64 | -16 | 15 | 39 | 41 | 263.16 | 75.66 | -153.10 | 4 | SS | 123 |
| C1_4 | 19 | 77 | 16 | 6 | 21 | 64 | 285.31 | 74.42 | 166.49 | 4 | SS | 153 |
| C2_1 | 348 | 29 | -29 | 15 | 10 | 18 | 316.27 | 51.98 | -109.38 | 1 | NF | 110 |
| C2_2 | 337 | 58 | -26 | 6 | 14 | 5 | 123.52 | 87.05 | -169.99 | 1 | NS | 117 |
| C2_3 | 336 | 50 | -57 | 18 | 9 | 23 | 81.49 | 68.18 | -145.19 | 2 | NF | 133 |
| C3_1 | 193 | 74 | -20 | 5 | 30 | 26 | 103.86 | 76.41 | -115.86 | 2 | SS | 151 |
| C4_1 | 33 | 80 | -3 | 4 | 11 | 18 | 288.73 | 70.81 | -163.03 | 1 | SS | 169 |
| C4_2 | 166 | 42 | -67 | 8 | 5 | 13 | 127.32 | 88.12 | 159.99 | 1 | NF | 148 |
| C5_1 | 5 | 64 | -24 | 17 | 36 | 39 | 106.04 | 68.56 | -151.90 | 3 | SS | 145 |
| C6_1 | 180 | 80 | -18 | 9 | 15 | 54 | 110.71 | 50.02 | -122.99 | 4 | SS | 136 |
| C6_2 | 218 | 70 | 2 | 14 | 24 | 34 | 273.23 | 72.28 | -169.50 | 4 | SS | 174 |

**Table 3 Selected fault plane solution for each cluster/event with corresponding uncertainty range, auxiliary fault plane, quality factor (Table A2), corresponding stress regime (see Sect. 4.5) and orientation of $S_{Hmax}$.**

**4.5 Direction of Maximum Horizontal Stress $S_{Hmax}$**

From our focal mechanisms we determine the direction of the maximum horizontal stress $S_{Hmax}$ in our research area. For this
analysis we use the same conventions as used for focal mechanisms which are included in the World Stress Map (WSM, Zoback, 1992, Heidbach et al., 2016). We assign a stress regime (strike-slip (SS), normal-faulting (NF), transverse faulting (TF), and mixed ones NS, TS) depending on the plunge of the pressure- (P), tension- (T), and B-axis (Zoback, 1992). Depending on the stress regime the direction of $S_{Hmax}$ is either the P- or B-axis trend or the T-axis trend plus 90° (Table 3, for details see Zoback, 1992). This worked well for all clusters with the exception of cluster C1_1. For this cluster we are not able
to derive a stress regime as all three axes have a moderate plunge in a range of 25° to 45°. In this case it is not possible to differ clearly between a maximum and minimum stress component (Zoback, 1992). For this reason, the P-axis trend is taken as $S_{Hmax}$



direction of cluster C1_1 and it must be considered with care. All calculated $S_{Hmax}$ directions are plotted together with the WSM data for our wider research area in Fig. 8. As most of our earthquakes have *ML* smaller 2.5, the rock volume involved in the earthquake rupture is quite small. For this reason, we average the calculated trend of $S_{Hmax}$ of all clusters to get one $S_{Hmax}$

direction estimate for the area of the ASZ. This seems to be reasonable as in comparison with the WSM we find only minor deviations of the $S_{Hmax}$ directions (Fig. 8). Furthermore, Rivera and Cisternas (1990) or Xu et al. (1992) demonstrated that composite solutions of stress field estimates from small earthquakes in a defined area represent a reliable regional stress field estimation. The median of the trend of $S_{Hmax}$ is 147° ± 31° at the ASZ. We use the 95% confidence range to give an error estimate. This results in an error of ± 31°. For later comparison we also calculated σ of $S_{Hmax}$, which is 20°.

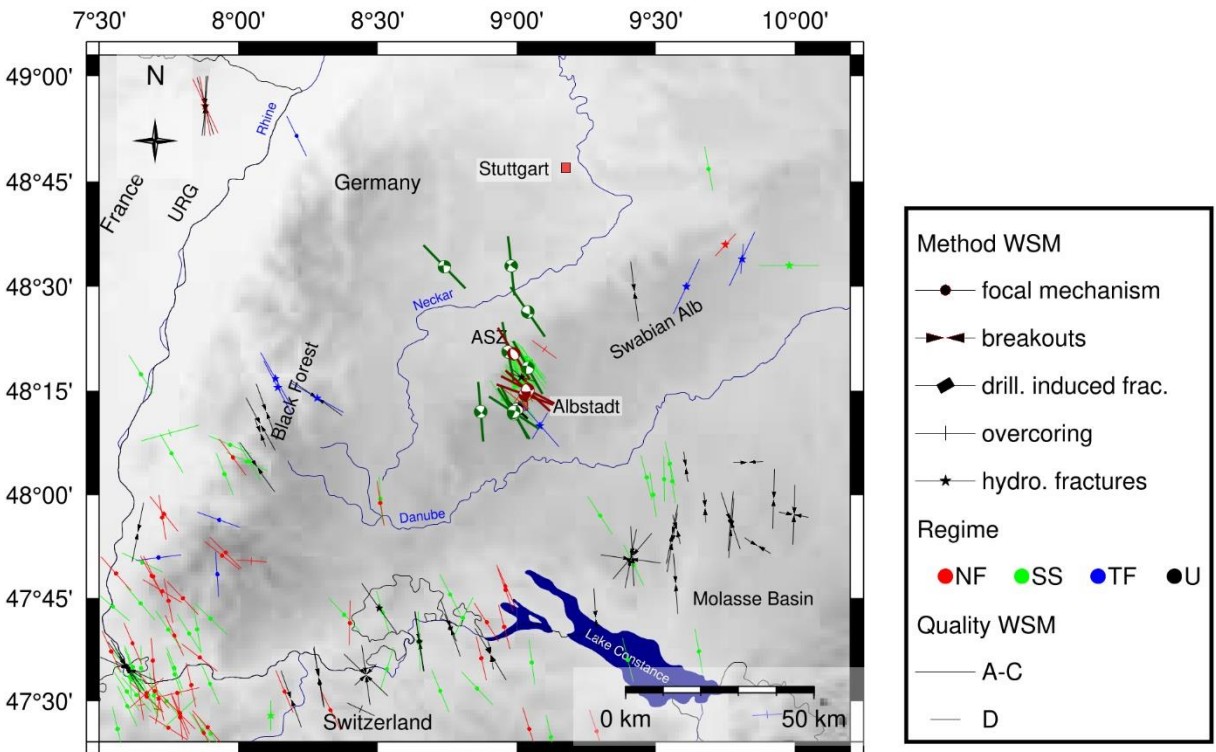


**Fig. 8 Orientation of $S_{Hmax}$ in the research area after the World Stress Map (WSM, light colors, Heidbach et al., 2016) and our new focal mechanisms and their corresponding $S_{Hmax}$ (dark colors, thick lines), color-coded by stress regime (NF=normal faulting, SS=strike-slip, TF=transverse faulting, U=unknown). The WSM data is scaled by quality. Quality A-C indicates a maximum error estimate of 25° (the good determined data), quality D of 40°. Topography is based on the ETOPO1 Global Relief Model (Amante**

**and Eakins, 2009).**




# 5 Results & Discussion

## 5.1 Velocity model and station delay times

The finally selected minimum 1-D seismic velocity model ASZmod1 consists of 5 layers (Fig. 4a and b). The layer boundaries

are based on the seismic refraction interpretation of Gajewski et al. (1987). Layers with very similar seismic velocities were combined during the inversion process to keep the model as simple as possible (Occam's principle). The determined seismic velocities increase with depth and they are well constrained between 2.5 km and 18.25 km depth (Table 1). The layers between -2 km to 2.5 km depth are not very stable due to the non-uniform distribution of rays and sources. Below 18.25 km depth we also have low resolution as all events used for inversion occur above. The comparison with the LED models gives a good

agreement with both, the Swabian Jura and the Baden-Württemberg models (Fig. 4a). Our layer between 2.5 km and 5.6 km depth is in good agreement with the Swabian Jura model, whereas the deeper layer has a higher agreement with the Baden-Württemberg model (Fig 4a). The Swabian Jura model has a finer layering for the uppermost 2 km. We also used the Swabian Jura model as input model for inversion, but due to the short horizontal raylength in comparison with the vertical raylength and the lack of events in the uppermost layers, the random seismic velocity starting models did not converge in the uppermost

layers (Fig. 3), therefore we chose the very simple layering.

The $vp/vs$-ratio is between 1.67 and 1.75 for all layers and it decreases with depth. In comparison, the LED uses a constant $vp/vs$-ratio of 1.72 for Baden-Württemberg and 1.68 for the Swabian Alb, which agrees with our overall observed $vp/vs$-ratio (Fig. 4b, Bulletin-Files des Landeserdbebendienstes B-W, 2018). The higher $vp/vs$-ratio of 1.75 in the first layer is a result of the manually fixed seismic velocities during the inversion process. In the second layer the $vp/vs$-ratio is also 1.75 which may

be caused by the numerical instability of this layer and should be interpreted with care. In our best determined layers (layer 3 and 4) our model has similar $vp/vs$-ratios as the Swabian Jura model of the LED (Fig. 4b, Bulletin-Files des Landeserdbebendienstes B-W, 2018).

The station delay times of the P- and S-waves have a simple pattern of increasing delay times with distance to reference station MSS (Fig. 6). Their very low values in the area of the ASZ demonstrate that the $vp$ and $vs$ distributions of ASZmod1 very well

represent the true seismic velocities in this area. Around the ASZ, the central Swabian Alb and the Molasse Basin are characterized by positive station delay times, thus slower seismic velocities along the propagation paths relative to ASZmod1. Other areas like the Black Forest exhibit negative delay times, so faster seismic velocities than ASZmod1.

The lateral seismic velocity contrasts of the different near-surface layers of Baden-Württemberg are small in comparison with our station delay times. For this reason, we compare our station delay times with the lateral depth variations of the crystalline

basement to find a possible relationship. The basement depth is described by the 3-D geological model of the Geological Survey of Baden-Württemberg (Rupf and Nitsch, 2008). Based on this model we estimate the vertical travel time at all our recording stations with more than either 5 P- or S-phase travel time picks using the seismic velocities of the first layer in ASZmod1 from the basement top to each recording station. For these values we calculated the travel time differences of all stations relative to station MSS and compared the results (Fig. 9) with our real station delay times (Fig. 6). As result we find




that the calculated travel time differences due to basement depth variations correlate to more than 85% with our station delay

times. Hence, basement depth variations are the main reason for the observed station delay times in our study region. The

remaining 15% of the station delay time terms may be explained by non-vertical ray path effects and lateral variations in

seismic velocity due to different near-surface rock types. Furthermore, other lateral heterogeneities like dipping or wave

guiding layers may influence the station delay times as well.

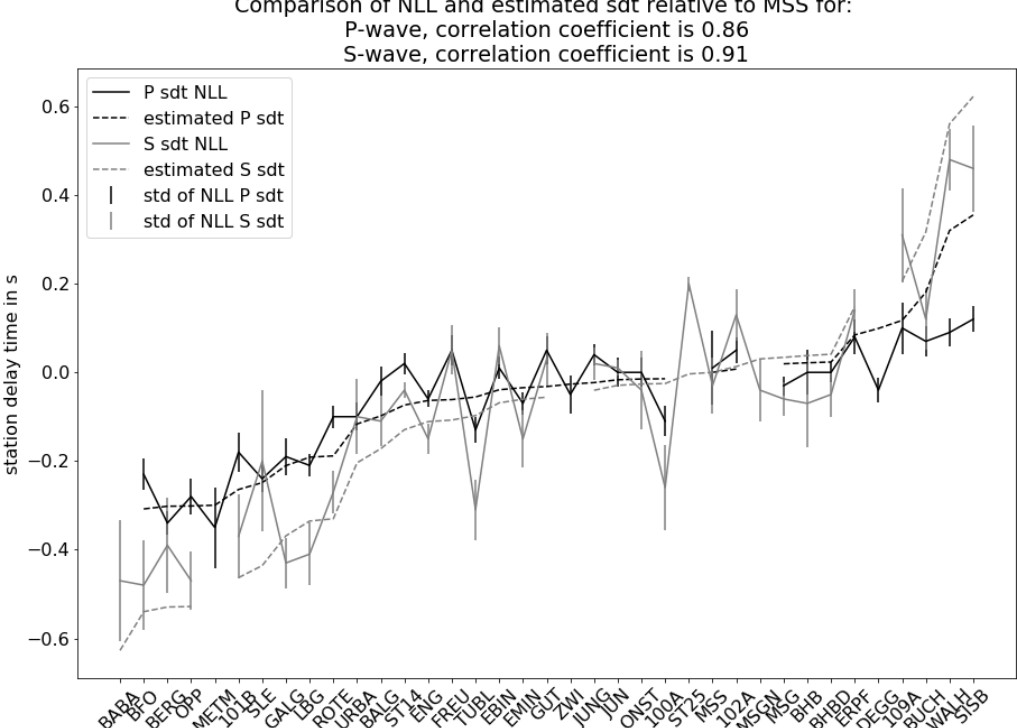


**Fig. 9 Comparison of NLL station delay times (sdt) and estimated station delay times due to depth variations of the crystalline basement. P-waves (black) and S-waves (gray). Stations along the x-axis are sorted from shallow to deep crystalline basement model depth.**






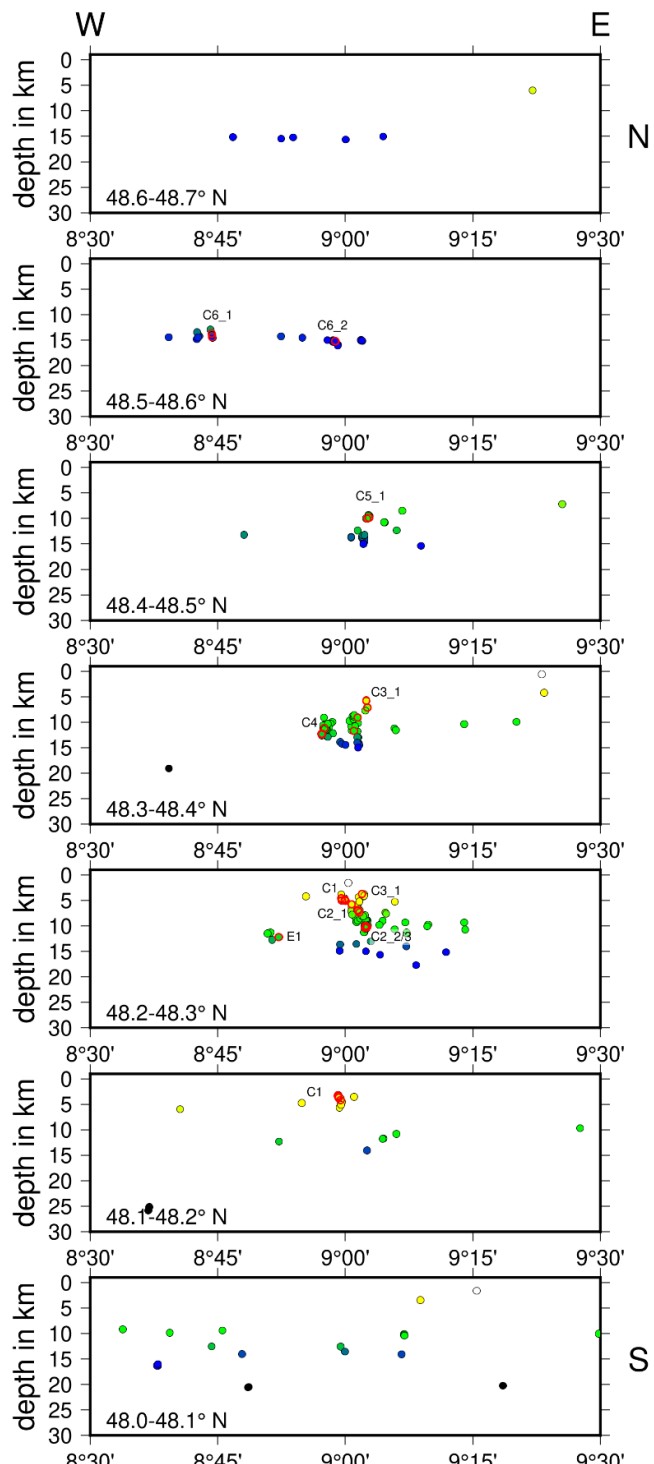

**Fig. 10 Seismicity distribution of the ASZ from north (top) to south (bottom). Circles indicate hypocenters in the corresponding slice, color-coded with depth (as in Fig. 7).**



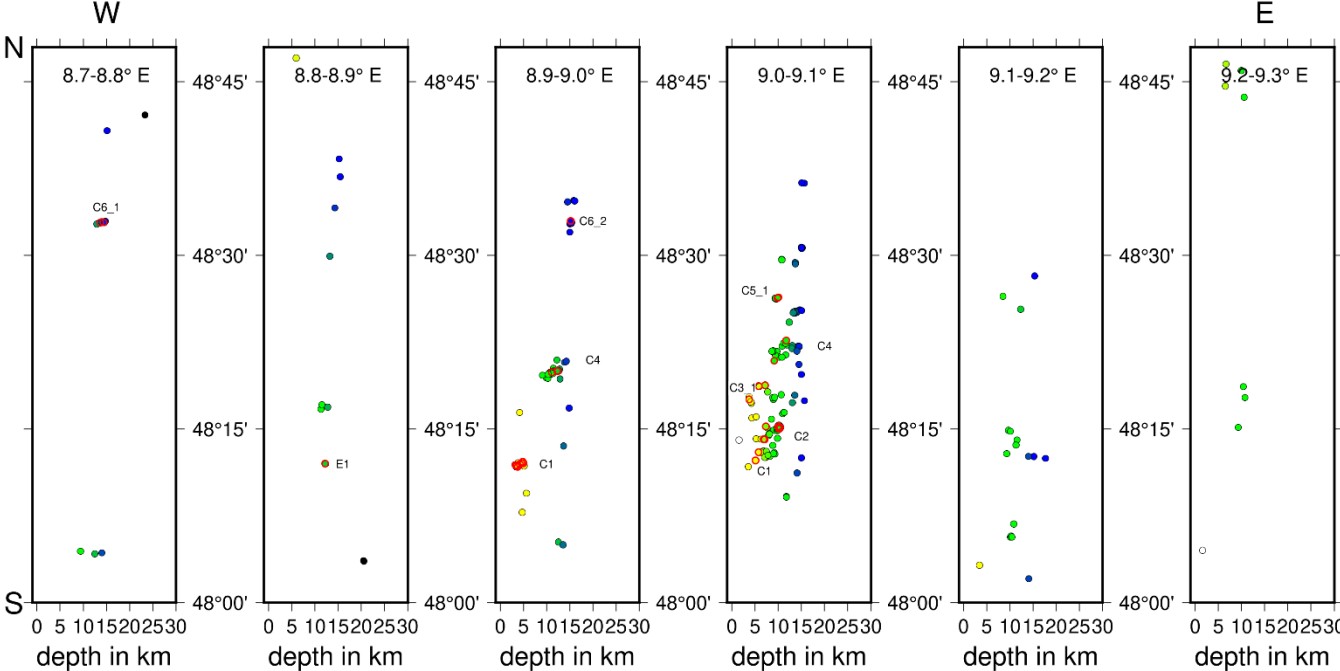

**Fig. 11 Seismicity distribution of the ASZ from west (left) to east (right). Circles indicate hypocenters in the corresponding slice, color-coded with depth (as in Fig. 7).**

**5.2 Seismicity and fault plane solutions of the ASZ**

The seismicity of the ASZ (Fig. 7) aligns almost north-south. Our relocated earthquakes occur in a depth range of 1 km to 18 km. If we follow the seismicity distribution from south to north, the minimum hypocenter depth increases from around 3 km to 5-14 km. Earthquakes below 18 km depth are rare at the ASZ. The top of the lower crust is at about 18-20 km depth (Gajewski and Prodehl, 1985, Aichroth et al., 1992), therefore, seismicity is concentrated in the upper crust. The hypocenters can be separated in several fault segments. This segmentation gets more obvious if we analyze east-west and north-south slices

(Fig. 10 & 11). In the north of the river Neckar (48.5-48.7° N), mainly deep (around 15 km depth) earthquakes occur, which can be separated into two clusters, one at 8.75 °E (C6_1) and one at 8.95 °E (C6_2, Fig. 10). Between the river Neckar and the town of Hechingen (48.3-48.5° N) we observe seismicity in the depth range of 5-15 km. There are three separate clusters, one west of 9 °E, directly south of Hechingen (C4), and two east of 9 °E (C5_1 and C3_1). Near the town Albstadt (48.2-48.3° N) the seismicity occurs across the whole seismically active depth range (1.5–18 km). Most seismicity happens between 9 °E and





9.1 °E (C2, C3_1). In 2 km to 8 km depth we find a small seismicity cluster southwest of Albstadt (8.9-9.0° E, C1). This cluster can be traced southward to 48.2 °N (48.1-48.2° N, C1).

Most of the fault plane solutions are characterized by the typical NNE-SSW strike of the ASZ, but we also observe some events with NNW-SSE strike (Fig. 7). Furthermore, most of the dip angles are greater than 60°, indicating a steep fault plane (Fig.12). If the rake angle is close to zero it indicates sinistral strike-slip faulting, if it is close to -180° or 180° it indicates

dextral strike slip faulting. Negative rake angles smaller than zero and greater than -180° imply normal faulting components, whereas positive rake angles imply reverse faulting components. Here, the events with a lower dip angle have a higher angle in rake, hinting to normal faulting components for negative rake angles and reverse faulting components for positive rake angles. The events with a rake close to zero indicate sinistral strike-slip components, which is the typical faulting mechanism of the ASZ (e.g. Turnovski, 1981, Stange and Brüstle, 2005). Comparing the fault plane solutions determined by Turnovski

(1981) and the fault plane solution of the 22nd March 2003 earthquake (Stange and Brüstle, 2005) with our newly determined fault plane solutions we observe very similar dominating faulting mechanisms (Fig. 12).

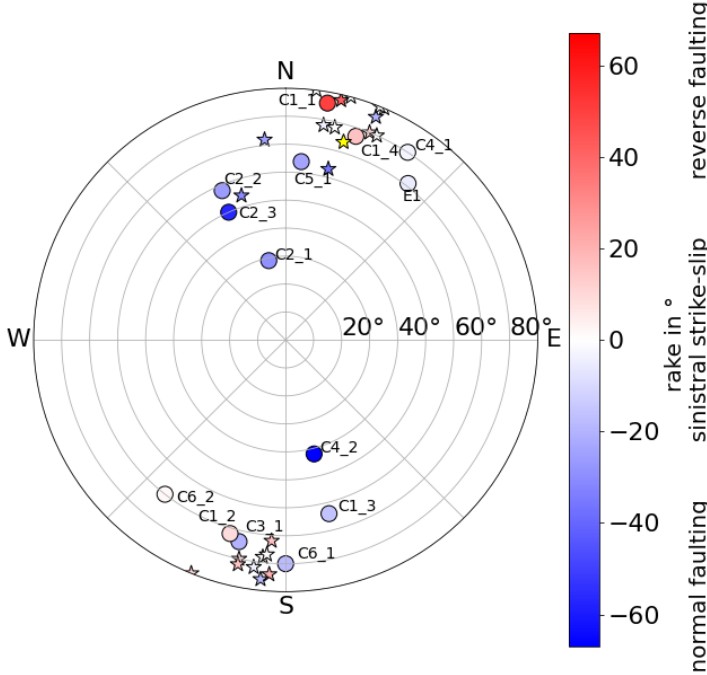

**Fig. 12 Strike, dip, and rake of all measured fault plane solutions with corresponding cluster code. Yellow star represents strike and dip of the 22nd March 2003 earthquake (Stange and Brüstle, 2005). Other stars represent fault plane solutions calculated by**
**Turnovsky (1981) for the earthquake series in 1978. Negative rake angles hint to normal faulting components and positive to reverse faulting components. Events with a rake close to zero exhibit sinistral strike-slip components. Colorbar is scaled by minimum and maximum rake angle.**





The normal faulting events (cluster C2, C4_2) have a strike of around 160°, almost parallel to the around NW-SE striking

HZG, and they are mainly located close to the boundary faults of the HZG (Fig. 7). The depth extension of the HZG is not well known but estimated from its extensional width and its dip angles of the main boundary faults at the surface. Based on these parameters, the boundary faults are thought to converge in about 2-3 km depth (Schädel, 1976). To test if the HZG boundary faults may relate to the events in 7-11 km depth we projected the faults of the fault plane solutions (both fault plane and auxiliary plane, Table 3) to the surface (Fig. S6). For most of our fault plane solutions we cannot differ clearly if the fault

plane or the auxiliary plane is the active fault plane as we only have few events available. In case of our cluster C2_2 we can clearly see a lineation of the earthquake hypcenters of the swarm in October 2018 along the strike with the fault plane in NNW direction. Nevertheless, this projected fault plane does not correlate with the strike of the HZG but is slightly oblique to it. The cluster C4 close to Hechingen has a higher correlation of the auxiliary fault planes with the strike of the HZG if the faults are projected for each event separately within the cluster to the surface (Fig. S7, auxiliary fault planes: C4_1, strike 289°, C4_2,

strike 127°, HZG, strike 130°, Baumann, 1986). If the auxiliary fault planes were the active fault planes, then cluster C4_1 would hint to dextral strike-slip movement und cluster C4_2 to reverse faulting at the northern HZG boundary fault. This faulting pattern may indicate that the HZG boundary faults reach to greater depth as already suggested by Schädel (1976) or Illies (1982). However, we cannot clearly distinguish between the two possible fault planes as the seismicity is still distributed too diffuse.

**5.3 Stress field around the ASZ**

From our fault plane solutions, we estimate the averaged direction of $S_{Hmax}$ to be 147°± 31° (σ = 20°). The orientation of $S_{Hmax}$ for southwest Germany is estimated to be around 150° with a σ of 24° (Reinecker et al., 2010) and for whole Western Europe 145° with a σ of 26° (Müller et al., 1992), which are both in agreement with our local $S_{Hmax}$ orientation. Houlié et al. (2018) also observes a similar stress field in east Switzerland, southeast of our research area. Furthermore, we find also a good

coincidence with the single $S_{Hmax}$ orientations of our clusters in comparison with the WSM (Fig. 8). Reinecker et al. (2010) suggest the gravitational potential energy of the Alpine topography as main source of the local stress field because the stress field orientation in the northern Alpine foreland is always perpendicular to the Alpine front. Kastrup et al. (2004) also observe a change of stress field orientation with the Alpine front for the northern Alpine foreland in Switzerland. They explain the change of the orientation of the minimum horizontal stress $S_h$ parallel to the Alpine front with a perturbation of the regional

European stress field due to the indentation of the Adriatic Block. Müller et al. (1992) identify the plate driving forces as sources of the maximum compression in NW to NNW direction for whole western Europe, only perturbated by large geological structures like the Alps. As our study area is quite small, we cannot observe major lateral stress variations, however, the good coincidence with the regional stress field (Müller et al., 1992, Reinecker et al., 2010) is a strong indication that the driving tectonic forces of the seismicity of the ASZ are the regional plate driving forces combined with the Alpine topography. Small

scale stress perturbations and variations of faulting mechanisms (Fig. 7 and 8) may be due to local heterogeneities of crustal





material causing variations in rigidity or preexisting structures. These factors may also play a role in the segmentation of the ASZ which will be analyzed in more details in the next years.

## 6 Conclusion and Outlook

We used our newly complemented seismicity catalog to invert for a robust new minimum 1-D seismic velocity model with
station delay times for the ASZ region. These station delay times can be explained by the depth variation of the crystalline basement in the upper crust of Baden-Württemberg (Fig. 9). The relocated seismicity of the years 2011 to 2018 pictures the ASZ as a complex fault structure, with its current main active focus between the cities Albstadt and Tübingen on the Swabian Alb. The hypocenter error estimates clearly become smaller for events after 2016 due to the densified seismic station network of the LED and the complementing AASN stations. Thus, we expect another improvement and an increase in detectable events
from 2019 onwards due to our additionally installed StressTransfer stations (Fig. 1).
Most of the seismicity takes place in a N to S oriented band east of 9 °E (Fig. 7). A spatial clustering of events is found which may indicate separate fault planes. If such a separation can be verified in the future, this segmentation would limit the maximum size of earthquake rupture planes and its related hazard potential (Grünthal and the GSHAP Region 3 Working Group, 1999). Nevertheless, we find the shallow cluster C1 slightly separated to the west from the other events, as well as the deeper cluster
C4 near Hechingen. The clusters C2 and C4_2 are clear normal faulting events, which were so far not observed for the ASZ. A relation of the clusters C4_1 and C4_2 with a continuation of the HZG into the crystalline basement is possible, as indicated by our fault line projections. Ongoing work will determine relative locations for all events from 2016 and following years to obtains an even sharper image of the fault planes of the ASZ. We also continue complementing our catalog with new earthquakes and fault plane solutions after 2018.
The estimated $S_{Hmax}$ has a NNW-SSW trend. This is in good agreement with other studies (Müller et al., 1992, Kastrup et al., 2004, Reinecker et al., 2010, Houlié et al., 2018). As plausible driving forces of our local stress field, we identify the regional plate driving forces as well as the Alpine topography (Müller et al., 1992, Kastrup et al., 2004, Reinecker et al., 2010). To get a better estimate of the stress field around the ASZ we plan to invert our fault plane solutions to determine the whole stress field tensor. Furthermore, within the StressTransfer project similar investigations are planned for the URG to the west and the
Molasse Basin south-east of the ASZ, to get a better understanding of the stress field in the northern Alpine foreland of southwest Germany.



**Appendix**

| Pick          time uncertainties | lp-ep <= 0.05 s | 0.05 s < lp-ep <= 0.1 s | 0.1 s < lp-ep <= 0.2 s | 0.2 s < lp-ep <= 0.4 s | lp-ep > 0.4s |
|---|---|---|---|---|---|
| Quality | 0 | 1 | 2 | 3 | 4 |

**Table A1 Definition of the error quality relationship. lp-ep represents the time window in which the final pick is manually selected.**

|  | $\Delta x \le 10°$ | $10° < \Delta x <= 20°$ | $20° < \Delta x <= 30°$ | $30° < \Delta x <= 40°$ | $40° < \Delta x$ |
|---|---|---|---|---|---|
| Quality | 0 | 1 | 2 | 3 | 4 |

**Table A2 Classification of the qualities used for focal mechanisms. Δx represents Δstrike, Δdip and Δrake. The lowest quality of all three parameters is given to the fault plane solution.**

**Code availability**

The preprocessing and picking analyses of the data were done in Python with the open-source toolbox ObsPy (Beyreuther et al., 2010, https://github.com/obspy/obspy/wiki). For further data analyses we used the freely available programs VELEST (Kissling et al., 1994, Kissling et al., 1995, VELEST Version 4.5, https://seg.ethz.ch/software/velest.html), NonLinLoc (Lomax et al., 2000, http://alomax.free.fr/nlloc/index.html) and FocMec (Snoke 2003, http://ds.iris.edu/pub/programs/focmec/). All figures were created with either the matplotlib library in Python (https://matplotlib.org/) or the Generic Mapping Tools (GMT, Wessel et al., 2019, https://www.generic-mapping-tools.org/).

**Data availability**

We used the restricted Bulletin-Files of the state earthquake service of Baden-Württemberg (LED), which were provided to us by the LED (Bulletin-Files des Landeserdbebendienstes B-W, 2018, https://lgrb-bw.de/erdbeben/index_html?lang=1). We analyzed the data of four AlpArray stations of the Z3 network, which will become available on 1[st] April 2022 to people outside the AlpArray Working Group (http://www.alparray.ethz.ch/en/seismic_network/backbone/data-access/). The data of the StressTransfer Network is currently restricted and will become available on 1[st] April 2022.

**Team list**

The complete member list of the AlpArray Working Group can be found at http://www.alparray.ethz.ch.



**Author contribution**

Sarah Mader carried out the fieldwork, analyzed the data with self-written and open-source programs and prepared the manuscript and figures. Joachim Ritter and Klaus Reicherter formulated the project, obtained funding and started the fieldwork. All three were involved in writing the manuscript. The AlpArray Working Group organized and coordinated the AlpArray

Seismic Network operation.

**Competing interests**

The authors declare that they have no conflict of interest.

**Acknowledgments**

We thank the State Earthquake Service of Baden-Württemberg (Landeserdbebendienst B-W) for providing waveform data,
pick files and internal information (Az4784//18_3303). We thank Andrea Brüstle, Tobias Diehl, and Dietrich Lange for programming and program related support and helpful discussions. Waveform recordings were provided by the AlpArray Seismic Network (2015). Recording instruments were provided by the KArlsruhe BroadBand Array, Karlsruhe Institute of Technology. S.M. and J.R. were supported by the Deutsche Forschungsgemeinschaft (DFG) by grant RI1133/13-1 and K.R. by grant RE1361/31-1 within the framework of DFG Priority Programme "Mountain Building Processes in Four Dimensions
(MB-4D)" (SPP 2017). Felix Bögelspacher, Felix Pappert, and Werner Scherer assisted the fieldwork. Michael Frietsch helped with the data management. We thank numerous people and associations for supporting the installation of the mobile recording instruments on their land. We acknowledge support by the KIT-Publication Fund of the Karlsruhe Institute of Technology.
We thank the AlpArray Seismic Network Team: György HETÉNYI, Rafael ABREU, Ivo ALLEGRETTI, Maria-Theresia APOLONER, Coralie AUBERT, Simon BESANÇON, Maxime BÈS DE BERC, Götz BOKELMANN, Didier BRUNEL,
Marco CAPELLO, Martina ČARMAN, Adriano CAVALIERE, Jérôme CHÈZE, Claudio CHIARABBA, John CLINTON, Glenn COUGOULAT, Wayne C. CRAWFORD, Luigia CRISTIANO, Tibor CZIFRA, Ezio D'ALEMA, Stefania DANESI, Romuald DANIEL, Anke DANNOWSKI, Iva DASOVIĆ, Anne DESCHAMPS, Jean-Xavier DESSA, Cécile DOUBRE, Sven EGDORF, ETHZ-SED Electronics Lab, Tomislav FIKET, Kasper FISCHER, Wolfgang FRIEDERICH, Florian FUCHS, Sigward FUNKE, Domenico GIARDINI, Aladino GOVONI, Zoltán GRÁCZER, Gidera GRÖSCHL, Stefan HEIMERS, Ben
HEIT, Davorka HERAK, Marijan HERAK, Johann HUBER, Dejan JARIĆ, Petr JEDLIČKA, Yan JIA, Hélène JUND, Edi KISSLING, Stefan KLINGEN, Bernhard KLOTZ, Petr KOLÍNSKÝ, Heidrun KOPP, Michael KORN, Josef KOTEK, Lothar KÜHNE, Krešo KUK, Dietrich LANGE, Jürgen LOOS, Sara LOVATI, Deny MALENGROS, Lucia MARGHERITI, Christophe MARON, Xavier MARTIN, Marco MASSA, Francesco MAZZARINI, Thomas MEIER, Laurent MÉTRAL, Irene MOLINARI, Milena MORETTI, Anna NARDI, Jurij PAHOR, Anne PAUL, Catherine PÉQUEGNAT, Daniel PETERSEN,
Damiano PESARESI, Davide PICCININI, Claudia PIROMALLO, Thomas PLENEFISCH, Jaroslava PLOMEROVÁ, Silvia





PONDRELLI, Snježan PREVOLNIK, Roman RACINE, Marc RÉGNIER, Miriam REISS, Joachim RITTER, Georg RÜMPKER, Simone SALIMBENI, Marco SANTULIN, Werner SCHERER, Sven SCHIPPKUS, Detlef SCHULTE-KORTNACK, Vesna ŠIPKA, Stefano SOLARINO, Daniele SPALLAROSSA, Kathrin SPIEKER, Josip STIPČEVIĆ, Angelo STROLLO, Bálint SÜLE, Gyöngyvér SZANYI, Eszter SZŰCS, Christine THOMAS, Martin THORWART, Frederik

TILMANN, Stefan UEDING, Massimiliano VALLOCCHIA, Luděk VECSEY, René VOIGT, Joachim WASSERMANN, Zoltán WÉBER, Christian WEIDLE, Viktor WESZTERGOM, Gauthier WEYLAND, Stefan WIEMER, Felix WOLF, David WOLYNIEC, Thomas ZIEKE, Mladen ŽIVČIĆ and Helena ŽLEBČÍKOVÁ.

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
