# Peer review of "Seismicity and seismotectonics of the Albstadt Shear Zone in the northern Alpine foreland"

_Solid Earth, 2020_

## Referee Comment (RC1) · Anonymous Referee #1 · 27 Oct 2020

The paper, " Seismicity and seismotectonics of the Albstadt Shear Zone in the northern Alpine foreland" by Mader et al., presents a study of the seismic activity in an intraplate region affected by regular low and moderate seismicity in Germany. They take advantage of the recent densification of the regional seismic network, in the framework of both the AlpArray European project and a local project specifically focusing on this area. The paper is well written and both the methodology and the results are well introduced. I suggest some corrections to improve some parts of the manuscript, and I make below some comments that the authors should discuss in the text. I believe that this study is important to get a better perspective of the origin of the seismic activity in such regions which remains poorly understood.

Regarding the region under study, I would recommend to the authors to give more

informations on the initial catalog. First, some more details on the quality of the locations, especially at depth (fixed or not), and the magnitudes. Second, it is important to mention if the catalog is discriminated with natural and anthropic events, and how. This is a crucial point. The catalogue used in this study corresponds to the initial catalog of the LED, complemented by additional pickings. I understand that this would represent another significant work, but I found a bit regrettable to not used the dense temporary network to decrease the completeness magnitude and enlarge the catalogue to get a better spatial distribution of the seismicity with a lower magnitude. Using a simple detection approach and the semi-automatic manual picking would help to improve the dataset. In general, the discussion regarding the magnitudes is lacking (also in figure see Figs. 7) In Fig. 1, I am surprised to see so many stations with very few picks. In Figure 5 as well, the low number of picks (P+S) per event is striking taking into account the density of the network. I suggest to add the focal mechanisms with the polarities in the Suppl. Mat. Within the text (section 4.4), I would recommend to modify a bit around L. 311, because we first think that the authors are doing composite mechanisms. The authors should mention in the text if the phase amplitudes are also taken into account in the method they used to determine the nodal planes. In section 4.5, all the arguments to chose a strike-slip regime should be included here. I am not very convinced by the method followed to determined ShMax, which is mainly appropriated for large events. For micro-seismicity, it seems more relevant to consider the stress field in the area under study homogeneous and conduct a inversion of the whole set of focal mechanisms (e.g., Rivera and Cisternas, 1990; Julien and Cornet, 1989, Michael, 1984, 1984; HardebeckÂăand Michael, 2006). I am not sure that the spatial variations of the Shmax direction can be relevant here, as mentioned by the authors. The use of this large diversity of focal mechanisms probably helps to constrain a stress regime and stress direction. The part of the discussion dealing with the fault plane solution is unclear. Looking at the Fig 12, it seems that reverse slips occur mainly on the most dipping planes, which can be surprising, and the authors should explain what his the main point of this figure. I found that the last section of the discussion could have been

strengthened. After reading it, we do not see what is the contribution of this work to better understand the stress field of this region.

Figures: Figure 1: This figure should be improved by increasing the size of the maps and by adding the corresponding geological background (transparency on shaded topography) since it is quite developed within the texte (section 2). The epicenters are either marked by circles or gray dots, why? I would recommend as well to change the topographic background by adding an illumination on the slope gradients. Add a rectangle on (a) corresponding to the zone of (b). Add Lake Constance. The magnitude scale is not appropriate since the large events are indicated by stars. More structural and tectonic details on (b), it remains unclear. Figure 2: the limits are confusing. I would recommend to keep the same frames than Fig. 1a. Also, the same colors used in Fig. 1a for the stations are also confusing. Use also a shaded topography. Indicate the station MSS (the white triangle is not visible in Fig 6). Figure 7: the magnitude scale is required Figure 8: indicate the area of study and indicate differently your data. Figure 10 and 11 should be combined into only one.

Minor comments: L. 14: tone down this sentence by removing ÂńÂăonlyÂăÂż, some geophysical imaging and paleo-seismological studies could help as well. L. 45: major damage: what kind? L.47: its ÂńÂăgeometryÂăÂż. Unclear, if you mean geometry of the seismicity, this is not correct. I would change into ÂńÂăto derive the geometry, the segmentation of the faulting patternÂăÂż. L. 49: you mean ÂńÂăprior large earthquakesÂăÂż? L. 51: add 'temporary' L. 58: remove minimum here, this is confusing for the Introduction L. 59: change geometry of the ASZ into ÂńÂăthe geometry of the fault pattern at depth in the ASZÂăÂż L. 60: remove ÂńÂăpermanentÂăÂż L. 84: It is not clear if it is the current or past stress field ? L. 94: The direction of the current principal stresses are constant, the amplitudes are not constrained. L. 104: ÂńÂăen echelon features compensating the displacements of the ASZÂăÂż: I am not sure to understand. L.112: ÂńÂămoderate displacementsÂăÂż : recent, Quaternary? L. 115: Add ÂńÂă(Fig. 1a)ÂăÂż after Lake Constance. L. 116: change ÂńÂălimitÂăÂż into

ÂńÂălimitedÂăÂż L. 116 to 119: these lines should be reformulated, they are unclear. L. 127, 173, 435 + caption_Fig12: remove ÂńÂăthÂăÂż and ÂńÂăndÂăÂż to the dates. L. 128: add ÂńÂărespectivelyÂăÂż. L. 128-129: ÂńÂăThe average. . .ÂăÂż. This is co-seismic slip rates. Mention from which data they are deduced. The same for the return period, mention briefly how this is estimated. L. 132: change ÂńÂăin a depthÂăÂż into ÂńÂăat a depthÂăÂż L. 134: the term ÂńÂăextensionÂăÂż is confusing here (lateral extend. . .) L. 142: ÂńÂăat the beginning of the summerÂăÂż. L. 158-159: Clarify ÂńÂăthe error boundaries are checkedÂăÂż. L. 180-181: give more explanations. L. 206 : ÂńÂăa few. . .ÂăÂż L. 222: ÂńÂălayersÂăÂż L. 226: ÂńÂăno eventÂăÂż L. 239: . . . the Stress Transfer stations from 2018ÂăÂż L. 367, 368: writhe the same number of decimals. L. 429-431: not necessary. And check consistency with the Fig 12, where rake varies between -65 and +65°, why?
* * *

---

## Referee Comment (RC2) · Thomas Plenefisch (Referee) · 4 Dec 2020

Comments to the paper: "Seismicity and seismotectonics of the Albstadt Shear Zone in the northern Alpine foreland"

by Sarah Mader, Joachim R. R. Ritter, Klaus Reicherter and the AlpArray Working Group

(Submitted to Solid Earth)

Reviewer: Thomas Plenefisch

The paper of Mader et al. is a classical seismicity and seismotectonic study. It focuses on the region of the Albstadt Shear Zone (ASZ), an area in SW Germany situated be-

tween the Black Forest and Rhine Graben in the west, the Alps in the south and the Franconian Jura in the northeast. From the seismicity point of you it is an intraplate area of moderate and permanently ongoing seismic activity. With an earthquake of magnitude close to 6 in 1978 the Albstadt shear zone comprises one of the two strongest earthquakes in Germany and its border regions over the last 50 years. Therefore, the region is also of great interest with regard to the assessment of seismic hazards in southern Germany.

The paper is divided in two parts. The first section is about the relocalization of the seismic events in the Albstadt Shear Zone from the time period 2011 to 2018 and the interpretation of the resulting seismicity pattern. Therefore, Mader et al. use the already existing catalog of phase picks of the state earthquake service of Baden-Württemberg and extend the dataset by phase picks of waveforms from the AlpArray network and from an project of the authors themselves, the so-called StressTransfer seismic network, which both have stations around the ASZ. The entire dataset of phase picks is then used to invert for a new minimum 1D model of P and S velocities as well as for station delay times. The final step is the relocalization of the earthquake dataset with the NonLinLoc program by using the new 5-layered velocity model. In the second part polarities of P phases and in some cases also for SH phases are used to invert for the focal mechanisms. Therefore, Mader et al. apply the FOCMEC-program. Since the magnitudes of some events are relatively small and often the signal-to-noise ratio is low the authors build clusters of events of narrow spaced hypocenters. All picks of one cluster are then used together in the inversion, quasi like a composite fault plane solution. In this way they calculate focal mechanisms for 36 earthquakes. However, do to the cluster approach this procedure results actually in 14 independent solutions.

The two subjects of the paper, firstly the relocation of earthquakes in an relatively small, intraplate area, the Albstadt shear zone in the Swabian Jura, with the aim to better resolve individual parts of the entire fault zone and secondly the focal mechanisms determination and its interpretation in the seismotectonic and geological context are of

general interest for seismologist, but also for a broader geoscientific community and thereby an interesting topic for Solid Earth.

The paper is clearly written and well structured. The input data seems to be of good quality and the applied techniques as well as the results sound to be reasonable for me in most parts. The first part of the paper comprising the calculation of the minimum 1D model as well as the relocalization is completely convincing me, I have no idea of any criticism. However, I have not so good feelings with the second part of the paper that deals with the focal mechanism and stress field determination. From my point of view therein there are some points which deserve an improvement or some more clarification respectively:

1. Focal mechanisms and stress field in SW Germany: When talking about stress field in SW Germany (line 83 – 86) it would be appropriate from my point of view to also reference the paper by Plenefisch & Bonjer (1997) and Bonjer (1997) who inferred the stress field of the Southern Rhine graben area by the inversion of 40 focal mechanisms determined by themselves (Bonjer 1997).

2. Presentation of the FOCMEC results: I wonder why the authors do not show any figures with the original output of the FOCMEC program, this means stereographic plots with used polarities and the calculated fault planes that are in accordance with the polarities. This is usual practice and gives the reader an impression about the distribution of the input data and the resulting and suitable pairs of fault planes. I suggest here to show a figure with the results for at least two or three earthquakes with different quality factors (after Table 3). The solutions for the other events or clusters respectively could be given either in the supplements or event better - since altogether these are 'only' 14 solutions - in one comprehensive figure in the paper itself. I think this is a must. I am aware of the fact that figure S5 shows the uncertainties in strike, dip and rake. This is a nice figure, but it does not replace the figure I proposed above.

3. Relative weighting and unity weighting: Please, describe shortly in the text, what is

the difference between these two weighting approaches.

4. SH polarities: In line 306 the authors state that in some cases they could determine SH polarities. After all these are 27 events according to Table 3. Since these SH polarities could be determined I assume that one could also determine amplitude ratios of SH/P and use them as input in the FOCMEC inversion. The use of amplitude ratios is scheduled in FOCMEC. It puts further constraints on the solutions and provides an easy way to determine one single solution by minimizing the differences between measured and calculated ratios. Why didn't you use such ratios, please comment on this topic. Perhaps the use of amplitude ratios could help in case of a small number of polarity observations and could made cluster analysis needless.

5. Missing a real inversion for the stress tensor: P-, T-and B-axis of a focal mechanism represent the strain axes. Only in case of a new fracture the strain axes automatically represent the principal stress axes. However, in case of a preexisting zone of weakness, which is commonly assumed for small size earthquakes, P-, T- and B-axes do not usually represent the principal stress axes. It is only the direction of the slip vector which is constrained by the orientation of the principal stress axes and the relative stress magnitude. An inversion of an ensemble of slip vectors finally allows the determination of the stress field and the strike of minimum or maximum horizontal stress axis respectively. I am not really convinced by the way Mader et al. determine their orientations of SHmax. I wonder why they have not performed a 'real' stress inversion of their focal mechanisms. There are several programs to do this (e.g. Gephart & Forsyth, 1984; Rivera and Cisternas, 1990; Michael, 1984; Hardebeck & Michael, 2006). I think this should be done before digging deeper into the interpretation of the stress field in and around the ASZ, even though I assume that the results will not dramatically change. The diversity of the focal mechanisms of the individual clusters which is obvious from Fig. 7 and Table 3 fulfills one essential requirement of a successful stress tensor inversion.

From my point of view the paper clearly deserves publication in Solid Earth after some

minor revisions or comments which I have listed above.

Please also note the supplement to this comment:
https://se.copernicus.org/preprints/se-2020-167/se-2020-167-RC2-supplement.pdf

---

## Author Comment (AC1) · 21 Dec 2020

**Answer to Anonymous Refree #1, author answers are written in blue:**

**Anonymous Referee #1**

The paper, " Seismicity and seismotectonics of the Albstadt Shear Zone in the northern Alpine foreland" by Mader et al., presents a study of the seismic activity in an intraplate region affected by regular low and moderate seismicity in Germany. They take advantage of the recent densification of the regional seismic network, in the framework of both the AlpArray European project and a local project specifically focusing on this area. The paper is well written and both the methodology and the results are well introduced. I suggest some corrections to improve some parts of the manuscript, and I make below some comments that the authors should discuss in the text. I believe that this study is important to get a better perspective of the origin of the seismic activity in such regions which remains poorly understood.

Regarding the region under study, I would recommend to the authors to give more informations on the initial catalog. First, some more details on the quality of the locations, especially at depth (fixed or not), and the magnitudes. Second, it is important to mention if the catalog is discriminated with natural and anthropic events, and how. This is a crucial point.

We added more information about the used catalog in section 3, line 140:

As basis for our study we use the earthquake catalog of the LED from 2011 to 2018 for earthquakes within the area close to the ASZ (8.5-9.5° E, 48-48.8° N, Fig. 1b). For these 575 earthquakes we received the bulletin files of the LED (Bulletin-Files des Landeserdbebendienstes B-W, 2018), consisting of hypocenter location, origin time, local magnitude ML and all phase travel time picks with corresponding quality and P-phase polarity. The LED picks from 2011 to 2018 are from 51 LED seismic stations and 14 seismic stations run by other agencies like the state earthquake service of Switzerland (Fig. 1a). Locations are determined with HYPOPLUS, a Hypoinverse variant (Stange and Brüstle 2005) which allows the usage of a 1.5-D seismic velocity model approach. Most hypocenter depths are calculated, around 9.7% of the depth are manually fixed. The median uncertainty for longitude, latitude and depth within the catalog are: 0.5 km, 0.6 km, and 2.0 km. The magnitude ML ranges from 0.0 to 3.4 with average uncertainties of about ±0.2 and the magnitude of completeness is around ML 0.6 (see Figure to be included in the appendix). The used catalog only contains natural events, as quarry blasts are sorted out and induced events do not occur in the study region.

[Figure]

The catalogue used in this study corresponds to the initial catalog of the LED, complemented by additional pickings. I understand that this would represent another significant work, but I found a bit regrettable to not used the dense temporary network to decrease the completeness magnitude and enlarge the catalogue to get a better spatial distribution of the seismicity with a lower magnitude. Using a simple detection approach and the semi-automatic manual picking would help to improve the dataset.

The detection of small magnitude earthquakes via template matching is current work in progress including location, uncertainty estimation and possible fault plane solution determination. When we did the study presented here, most of our StressTransfer seismic stations only run for a few months in 2018 and we didn't have access to the waveform data of the LED seismic stations yet.

Added hint in section 6, line 484:
Thus, we expect another improvement and an increase in detectable events from 2019 onwards due to our additionally installed StressTransfer stations (Fig. 1). Future work will take advantage of the densified seismic station network and focus on small magnitude event detection based on template matching in the area of the ASZ.

In general, the discussion regarding the magnitudes is lacking (also in figure see Figs. 7).

The magnitudes are taken from the LED catalog, these are well determined and cannot be improved within the current uncertainties (±0,2 units). At present we only use magnitude for scaling the size of the fault plane solutions (Fig. 7) and epicenters with magnitude (Fig. 1). As we only took them without changing or complementing anything we don't think a discussion is missing.

In Fig. 1, I am surprised to see so many stations with very few picks.

In Fig. 1 the colors represent the seismic station networks as stated by the legend entries, so permanent (black), StressTransfer (green) or AASN (yellow) and not the number of picks. Maybe you mean Fig. 2? Here, there are many stations with few picks, as Fig. 2 shows the data base used for vp and vs inversion and not the full catalog. The catalog for vp and vs inversion only contains high quality picks, as described in section 4.2 line 174ff. We clarify this in the figure caption of Fig. 2: Seismic stations are indicated as triangles and color-coded with the number of **high quality** picks at a station **used for the vp and vs inversion**.

In Figure 5 as well, the low number of picks (P+S) per event is striking taking into account the density of the network.

The network shown in Fig 1a is the full network which developed and changed with time. All events with yellow colors have 24 and more picks, as indicated by the small triangle on the right side at the color bar.

I suggest to add the focal mechanisms with the polarities in the Suppl. Mat.

Such a figure will be included.

Within the text (section 4.4), I would recommend to modify a bit around L. 311, because we first think that the authors are doing composite mechanisms.

We are doing composite mechanisms, this is now better clarified in the text.
L.311: For this reason, we combined events with similar hypocenter coordinates and polarity picks to determine a joint or composite solution (Table 2). In this way we combined them to clusters with a corresponding cluster code (Table 2).

The authors should mention in the text if the phase amplitudes are also taken into account in the method they used to determine the nodal planes.

No, at the moment they are not included, but we will include SH/P ratios for the revised version of the manuscript. (this was also suggested by reviewer 2)

In section 4.5, all the arguments to chose a strike-slip regime should be included here.

We took the same criteria for choosing a stress regime for each fault plane solution as described in detail in Zoback 1992, Table 3. Depending on the plunge (pl) of the P-, T- and B-axis of a fault plane solution, a specific stress regime is chosen:

| P | B | T | Regime |
|---|---|---|---|
| pl≥52° | | pl ≤ 35° | NF |
| 40° ≤ pl ≤ 52° | | pl ≤ 20° | NS |
| pl < 40° | pl ≥ 45° | pl ≤ 20° | SS |
| pl ≤ 20° | pl ≥ 45° | pl < 40° | SS |
| pl ≤ 20° | | 40° ≤ pl ≤ 52° | TS |
| pl ≤ 35° | | pl ≥ 52° | TF |

Taken from Zoback 1992, Table 3.

L. 341: We assign a stress regime (strike-slip (SS), normal-faulting (NF), transverse faulting (TF), and mixed ones NS, TS) depending on the plunge of the pressure- (*P*), tension- (*T*), and *B*-axis (Zoback, 1992). Depending on the stress regime the direction of $S_{Hmax}$ is either the *P*- or *B*-axis trend or the *T*-axis trend plus 90° (Table 3, for details see Zoback, 1992).

I am not very convinced by the method followed to determined ShMax, which is mainly appropriated for large events. For micro-seismicity, it seems more relevant to consider the stress field in the area under study homogeneous and conduct a inversion of the whole set of focal mechanisms (e.g., Rivera and Cisternas, 1990; Julien and Cornet, 1989, Michael, 1984, 1984; HardebeckĂˇaand Michael, 2006). I am not sure that the spatial variations of the Shmax direction can be relevant here, as mentioned by the authors. The use of this large diversity of focal mechanisms probably helps to constrain a stress regime and stress direction.

We understand your objections regarding the method we used to determine the direction of $S_{Hmax}$, as we mainly have small magnitude events. Originally, we planned to invert for the stress field in future work but we will now include the inversion in the revised version of the manuscript as suggested. Accordingly, section 4.5 will be reworked completely.

The part of the discussion dealing with the fault plane solution is unclear. Looking at the Fig 12, it seems that reverse slips occur mainly on the most dipping planes, which can be surprising, and the authors should explain what his the main point of this figure.

Changed:

Most of the fault plane solutions are characterized by the typical NNE-SSW strike of the ASZ, but we also observe some events with NNW-SSE strike (Fig. 7, Fig. 12). The events with a strike of NNE-

SSW are characterized by steep fault planes (dip angle greater 60°) and rake angles around 0°, hinting to sinistral strike-slip. This is the typical or main faulting mechanism of the ASZ (e.g. Turnovski, 1981, Stange and Brüstle, 2005). We also observe one fault plane solution with NNE-SSW strike with a clear reverse faulting component and a steep fault plane of 86° (cluster C1_1, Fig.12). The events with NNW-SSE strike have lower dip angles (smaller 60°) and negative rake angles, hinting to normal faulting (cluster C2, C4_2, Fig. 12). The here observed faulting behaviors can all be explained by a compressional stress regime with an average horizontal stress orientation of around 150° (Müller et al., 1992, Reinecker et al., 2010, Heidbach et al., 2016) acting on either the NNE-SSW or NNW-SSE oriented fault planes.

Comparing the fault plane solutions determined by Turnovski (1981) and the fault plane solution of the 22nd March 2003 earthquake (Stange and Brüstle, 2005, stars in Fig. 12) with our newly determined fault plane solutions we observe very similar dominating faulting mechanisms (strike slip, Fig. 12), but we also observe several normal faulting events with a strike of NNW-SSE. This is a new observation indicating the complexity of the fault zone.

I found that the last section of the discussion could have been strengthened. After reading it, we do not see what is the contribution of this work to better understand the stress field of this region.

We understand your point and will strengthen this section. This section will be changed into the interpretation of the stress field inversion which will be calculated for the revised manuscript.

Figures: Figure 1: This figure should be improved by increasing the size of the maps and by adding the corresponding geological background (transparency on shaded topography) since it is quite developed within the texte (section 2). The epicenters are either marked by circles or gray dots, why? I would recommend as well to change the topographic background by adding an illumination on the slope gradients. Add a rectangle on (a) corresponding to the zone of (b). Add Lake Constance. The magnitude scale is not appropriate since the large events are indicated by stars. More structural and tectonic details on (b), it remains unclear.

We understand your recommendations and will change the following points:
The map size will be increased. We will add a separate map with geological background and an illuminated topography (see figure below) as we think that Fig. 1a) is already very full and will become confusing. All events are shown by gray circles, scaled by magnitude, so dots are events where the circle is very small due to a small magnitude. The rectangle will be added in Fig. 1a) framing the area of Fig. 1b). Lake Constance is already included in the figure, but we will increase the font size for better visibility. The magnitude scale is shown for the catalog from 2011 to 2018, which is the catalog we will later use in our area. The large events shown as stars happened all before 2011 and are 'historic' events and are not included in the shown earthquake catalog. We will change the

figure caption to make it more clear. We show all known and assumed faults in the area of Fig. 1b) but only mark the relevant ones (HZG) and the tectonic features mentioned in the text (LG, SL).

**Fig. 1 a) Overview over our research area located in the southwest of Germany in the northern Alpine foreland. The ASZ is our research target. Black triangles represent permanent seismic stations of the LED and other agencies. Yellow triangles represent temporary AlpArray seismic stations. Green triangles display the fifteen temporary seismic stations of the StressTransfer network. The gray circles display the seismicity scaled by magnitude from 2011 to 2018. URG=Upper Rhine Graben**

**b) Close up of the area of the ASZ. Symbols like in Fig. 1a. Red framed triangle highlights central station Meßstetten (MSS) of the minimum 1-D seismic velocity model. White stars mark epicenters of the 4 strongest events with a magnitude greater than 5 in 1911, two in 1943 (same epicenter) and 1978, as well as the earthquake on 22nd March 2003 with a local magnitude of 4.4 (Leydecker, 2011), those events are not included in the earthquake catalog from 2011 to 2018 (gray circles scaled with magnitude like in Fig. 1a). White lines indicate known and assumed faults (Regierungspräsidium Freiburg, Landesamt für Geologie, Rohstoffe und Bergbau (Hrsg.), 2019). The Hohenzollern Graben (HZG) is the only tectonic feature in the close vicinity to the ASZ. Other large tectonic features of interest are the Lauchertgraben (LG) and the Swabian Line (SL).**

**Topography is based on the ETOPO1 Global Relief Model (Amante and Eakins, 2009).**

[Figure]

Figure 2: the limits are confusing. I would recommend to keep the same frames than Fig. 1a. Also, the same colors used in Fig. 1a for the stations are also confusing. Use also a shaded topography.

The limits are the same as Fig. 1a) for the area as well as for the topography colorbar. The meaning of the colors of the triangles is clearly stated by the colorbar and the figure caption.

Indicate the station MSS (the white triangle is not visible in Fig 6).
The size of the white triangle will be increased.

Figure 7: the magnitude scale is required
Will be included.

Figure 8: indicate the area of study and indicate differently your data.
Will be changed, also depending on the stress inversion results.

Figure 10 and 11 should be combined into only one.
Will be combined.

Minor comments:
L. 14: tone down this sentence by removing only, some geophysical imaging and paleo-seismological studies could help as well.
Changed:
As there are no visible surface ruptures, which may be connected to the fault zone, we study its characteristics by its seismicity distribution and faulting pattern.

L. 45: major damage: what kind?
Added:
Although the earthquakes caused major damage, like damage of buildings such as fractures in house walls and damaged roofs or chimneys, no surface ruptures have been found or described (e.g. Schneider, 1971).

L.47: its geometry. Unclear, if you mean geometry of the seismicity, this is not correct. I would change into to derive the geometry, the segmentation of the faulting pattern.
Changed:
For this reason, the ASZ can only be analyzed by its seismicity to derive the geometry, possible segmentation, and faulting pattern.

L. 49: you mean aprior large earthquakes?
Changed:

One of the best observed earthquakes happened on 22nd March 2003 and it was described as a sinistral strike-slip fault with a strike of 16° (Stange and Brüstle, 2005), which is similar to the observations from former events (e.g. Schneider, 1973, Turnovsky, 1981, Kunze, 1982).

L. 51: add 'temporary'

Changed:

Since 2005 the seismic station network of the state earthquake service of Baden-Württemberg (LED) was changed and extended (Stange, 2018) and in summer 2015 the installation of the temporary Alp Array Seismic Network (AASN) started (Hetényi et al., 2018).

L. 58: remove minimum here, this is confusing for the Introduction

Changed:

We calculate a new 1-D seismic velocity model and relocate the events.

L. 59: change geometry of the ASZ into the geometry of the fault pattern at depth in the ASZ

L. 60: remove permanent

Changed:

This procedure gives us a new view of the geometry of the fault pattern with depth of the ASZ based on its microseismic activity.

L. 84: It is not clear if it is the current or past stress field ?

Changed:

The current regional stress field of southwest Germany is dominated by an average horizontal stress orientation of 150° (e.g. Müller et al., 1992, Reinecker et al., 2010, Heidbach et al., 2016) and was determined from focal mechanism solutions, overcoring, borehole breakouts, and hydraulic fracturing (e.g. Kastrup et al., 2004, Reiter et al., 2015, Heidbach et al., 2016).

L. 94: The directions of the current principal stresses are constant, the amplitudes are not constrained.

We changed the sentence to:

The direction of $S_{Hmax}$ in our research area is quite constant, except of directly south (Albstadt-Truchtelfingen) and within (Albstadt-Onstmettingen) the HZG. There the $S_{Hmax}$ direction rotates about 20° counterclockwise into the strike direction of the HZG (130°, Baumann, 1986), which may be caused by a reduced marginal shear resistance.

L. 104: en echelon features compensating the displacements of the ASZ: I am not sure to understand.

With en echelon features we mean graben structures which open in a steplike arrangement at the surface above a partly decoupled strike-slip fault (see Reinecker and Schneider 2002 or Tron and Brun 1991). The tectonic model of Reinecker and Schneider (2002) connects the visible graben structures at the surface (like the HZG) with the ASZ.

Changed:

L 104: Reinecker and Schneider (2002) propose a tectonic model to relate the graben structures with the ASZ below. They apply the result of Tron and Brun (1991), who showed that the movement of a partly decoupled strike-slip fault in the subsurface can generate graben structures at the surface in a steplike arrangement. In the tectonic model the graben structures are the HZG, the Rottenburg Flexur, western Lake Constance, the Fildergraben, … (Reinecker and Schneider, 2002). The ASZ itself is the strike-slip fault, partly decoupled from the surface by a layer of evaporites of the salinar formation of the Middle Triassic from the overlying sedimentary layers (Reinecker and Schneider, 2002). Stange and Brüstle (2005) consider the bottom of the Mesozoic sediments as mechanical decoupling horizon as no earthquakes occur above 2 km depth.

L.112: moderate displacements : recent, Quaternary?

Sentence in paper: The faults in southwest Germany exhibit mainly moderate displacements. At the HZG, for example, the maximum vertical offset is of the order of 100 m to 150 m. The horizontal offset is considerably lower and more difficult to determine (Reicherter et al., 2008).

➔ We are searching for an age determination

L. 115: Add (Fig. 1a) after Lake Constance.

Changed:

Wetzel and Franzke (2003) identified a 5 km to 10 km broad zone of lineations along the 9 °E meridian striking predominantly N-S, NW–SW and ENE-WSE pursuable from Stuttgart to Lake Constance (Fig. 1a).

L. 116: change limit into limited

Changed:

The graben structures, like the HZG, are limited by the NW-SE striking faults (Reicherter et al., 2008).

L. 116 to 119: these lines should be reformulated, they are unclear.

Changed to:

Along the 9°E meridian Wetzel and Franzke (2003) identified a 5 km to 10 km broad zone of lineations pursuable from Stuttgart to Lake Constance (Fig. 1a). Those lineations strike predominantly N-S, NW-SW and ENE-WSW. The N-S and ENE-WSW striking faults limit the NW-SE striking graben structures, like the HZG (Reicherter et al., 2008). The NW-Se striking faults are expected to be

possibly active at intersections with N-S striking faults due to a reduction in shear resistance accompanied by aseismic creep (Schneider, 1979, Schneider, 1993, Wetzel and Franzke, 2003).

L. 127, 173, 435 + caption_Fig12: remove th and nd to the dates.
These are removed now.

L. 128: add respectively
Changed:
The average seismic dislocation rates along the ASZ are on the order of 0.1 mm/a, respectively (Schneider, 1993).

L. 128-129: The average: This is coseismic slip rates. Mention from which data they are deduced. The same for the return period, mention briefly how this is estimated.
Added:
The average seismic dislocation rates along the ASZ are on the order of 0.1 mm/a, respectively (Schneider, 1993). The return period of earthquakes along the ASZ with a magnitude of 5 is approximately 1000 years (Schneider, 1980, Reinecker and Schneider, 2002). Both estimates are based on historic earthquake records. → this will be clarified in the revised manuscipt

L. 132: change in a depth into at a depth
Changed:
From aftershock analyses and focal mechanism calculations we know that the ASZ is a steep NNE to SSW oriented sinistral strike-slip fault (e.g. Haessler et al., 1980, Turnovsky, 1981, Stange and Brüstle, 2005) in the crystalline basement, as all earthquakes occur at a depth greater than 2 km (Stange and Brüstle, 2005).

L. 134: the term extension is confusing here (lateral extend)
Changed to:
The lateral extend of the fault zone in N-S direction is still under debate:

L. 142: at the beginning of the summer.
Changed:
Additionally, within the AlpArray Project (Hetényi et al., 2018), nine seismic stations were installed starting in summer 2015 within 80 km distance to the ASZ, four of them directly around the ASZ (AlpArray Seismic Network, 2015, Fig. 1b).

L. 158-159: Clarify the error boundaries are checked.

Changed:

Finally, the uncertainty boundaries are checked by eye, and the phase pick is done manually between the two uncertainty boundaries.

L. 180-181: give more explanations.

We will update this point in the revised manuscript

L. 206 : a few: :

Changed:

The instability in both upper layers may be caused by a few refracting rays and therefore small horizontal raylengths through the layers as well as only a few earthquakes within these layers (Fig. S2c).

L. 222: layers

Changed:

The 2σ range is small for the third and fourth layers.

L. 226: no event

Changed:

The fifth layer also has larger 2σ uncertainties relative to layers three and four, which is caused by less ray coverage and no event located below 18.25 km depth.

L. 239: the Stress Transfer stations from 2018

Changed:

This reduction correlates well with the increased number of picks per event, so with the increased number of seismic stations around the ASZ due to the modification of the LED network and the installation of the AASN and the StressTransfer stations from 2018 (Hetényi et al., 2018, Stange, 2018, Fig. 5).

L. 367, 368: writhe the same number of decimals.

Changed:

The determined seismic velocities increase with depth and they are well constrained between 2.50 km and 18.25 km depth (Table 1). The layers between -2.0 km to 2.5 km depth are not very stable due to the non-uniform distribution of rays and sources.

L. 429-431: not necessary. And check consistency with the Fig 12, where rake varies between -65 and +65_, why?

L. 429-431 was omitted, see previous comment, rake angle in our case varies between -67° and 50° (see Table 3).

---

## Author Comment (AC2) · 21 Dec 2020

**Answer to Thomas Plenefisch (Referee), author answers are written in blue:**

**Thomas Plenefisch (Referee)**
thomas.plenefisch@bgr.de

Comments to the paper: "Seismicity and seismotectonics of the Albstadt Shear Zone
in the northern Alpine foreland"
by Sarah Mader, Joachim R. R. Ritter, Klaus Reicherter and the AlpArray Working
Group
(Submitted to Solid Earth)
Reviewer: Thomas Plenefisch
The paper of Mader et al. is a classical seismicity and seismotectonic study. It focuses
on the region of the Albstadt Shear Zone (ASZ), an area in SW Germany situated between the Black
Forest and Rhine Graben in the west, the Alps in the south and the
Franconian Jura in the northeast. From the seismicity point of you it is an intraplate area
of moderate and permanently ongoing seismic activity. With an earthquake of magnitude
close to 6 in 1978 the Albstadt shear zone comprises one of the two strongest
earthquakes in Germany and its border regions over the last 50 years. Therefore, the
region is also of great interest with regard to the assessment of seismic hazards in
southern Germany.
The paper is divided in two parts. The first section is about the relocalization of the seismic
events in the Albstadt Shear Zone from the time period 2011 to 2018 and the interpretation
of the resulting seismicity pattern. Therefore, Mader et al. use the already
existing catalog of phase picks of the state earthquake service of Baden-Württemberg
and extend the dataset by phase picks of waveforms from the AlpArray network and
from an project of the authors themselves, the so-called StressTransfer seismic network,
which both have stations around the ASZ. The entire dataset of phase picks is
then used to invert for a new minimum 1D model of P and S velocities as well as for
station delay times. The final step is the relocalization of the earthquake dataset with
the NonLinLoc program by using the new 5-layered velocity model. In the second part
polarities of P phases and in some cases also for SH phases are used to invert for
the focal mechanisms. Therefore, Mader et al. apply the FOCMEC-program. Since
the magnitudes of some events are relatively small and often the signal-to-noise ratio
is low the authors build clusters of events of narrow spaced hypocenters. All picks of
one cluster are then used together in the inversion, quasi like a composite fault plane
solution. In this way they calculate focal mechanisms for 36 earthquakes. However, do
to the cluster approach this procedure results actually in 14 independent solutions.
The two subjects of the paper, firstly the relocation of earthquakes in an relatively small,
intraplate area, the Albstadt shear zone in the Swabian Jura, with the aim to better
resolve individual parts of the entire fault zone and secondly the focal mechanisms
determination and its interpretation in the seismotectonic and geological context are of
general interest for seismologist, but also for a broader geoscientific community and
thereby an interesting topic for Solid Earth.
The paper is clearly written and well structured. The input data seems to be of good
quality and the applied techniques as well as the results sound to be reasonable for
me in most parts. The first part of the paper comprising the calculation of the minimum
1D model as well as the relocalization is completely convincing me, I have no idea of
any criticism. However, I have not so good feelings with the second part of the paper
that deals with the focal mechanism and stress field determination. From my point
of view therein there are some points which deserve an improvement or some more
clarification respectively:

1. Focal mechanisms and stress field in SW Germany: When talking about stress field in SW Germany (line 83 – 86) it would be appropriate from my point of view to also reference the paper by Plenefisch & Bonjer (1997) and Bonjer (1997) who inferred the stress field of the Southern Rhine graben area by the inversion of 40 focal mechanisms determined by themselves (Bonjer 1997).

We will include the mentioned papers.

Line 83ff: The regional stress field of southwest Germany is dominated by an average horizontal stress orientation of 150° (e.g. Müller et al., 1992, Plenefisch & Bonjer (1997), Reinecker et al., 2010, Heidbach et al., 2016) and was determined from focal mechanism solutions, overcoring, borehole breakouts and hydraulic fracturing (e.g. Kastrup et al., 2004, Bonjer (1997), Plenefisch & Bonjer (1997), Reiter et al., 2015, Heidbach et al., 2016). It is characterized by NW-SE horizontal compression and NE-SW extension (e.g. Kastrup et al., 2004) and developed during late Miocene (Becker 1993).

2. Presentation of the FOCMEC results: I wonder why the authors do not show any figures with the original output of the FOCMEC program, this means stereographic plots with used polarities and the calculated fault planes that are in accordance with the polarities. This is usual practice and gives the reader an impression about the distribution of the input data and the resulting and suitable pairs of fault planes. I suggest here to show a figure with the results for at least two or three earthquakes with different quality factors (after Table 3). The solutions for the other events or clusters respectively could be given either in the supplements or event better - since altogether these are 'only' 14 solutions - in one comprehensive figure in the paper itself. I think this is a must. I am aware of the fact that figure S5 shows the uncertainties in strike, dip and rake. This is a nice figure, but it does not replace the figure I proposed above.

Figures with polarities and fault plane solution fits will be included in the revised paper.

3. Relative weighting and unity weighting: Please, describe shortly in the text, what is the difference between these two weighting approaches.

Description added to line 320:
We compare both polarity weighting modes, relative weighting and unity weighting, of FOCMEC for all events, to explore if the result differs significantly (Snoke, 2003). In the unity weighting mode each wrong polarity counts as an error of one, whereas in the relative weighting mode polarity errors near a nodal plane count less than polarity errors in the middle of a quadrant, so they are weighted regarding their location in respect to the nodal planes. If we consider a wrong polarity pick close to a nodal plane, we would have to allow an error of one for unity weighting to get a solution whereas in the relative weighting mode an error of 0.15 would be sufficient.

4. SH polarities: In line 306 the authors state that in some cases they could determine SH polarities. After all these are 27 events according to Table 3. Since these SH polarities could be determined I assume that one could also determine amplitude ratios of SH/P and use them as input in the FOCMEC inversion. The use of amplitude ratios is scheduled in FOCMEC. It puts further constraints on the solutions and provides an easy way to determine one single solution by minimizing the differences between measured and calculated ratios. Why didn't you use such ratios, please comment on this topic. Perhaps the use of amplitude ratios could help in case of a small number of polarity observations and could made cluster analysis needless.

We only determined the polarities and didn't determine the amplitude values until now. But we agree with your recommendation and we will include the amplitude ratios in the revised version of the manuscript.

5. Missing a real inversion for the stress tensor: P-, T-and B-axis of a focal mechanism represent the strain axes. Only in case of a new fracture the strain axes automatically represent the principal stress axes. However, in case of a preexisting zone of weakness, which is commonly assumed for small size earthquakes, P-, T- and B-axes do not usually represent the principal stress axes. It is only the direction of the slip vector which is constrained by the orientation of the principal stress axes and the relative stress magnitude. An inversion of an ensemble of slip vectors finally allows the determination of the stress field and the strike of minimum or maximum horizontal stress axis respectively. I am not really convinced by the way Mader et al. determine their orientations of SHmax. I wonder why they have not performed a 'real' stress inversion of their focal mechanisms. There are several programs to do this (e.g. Gephart & Forsyth, 1984; Rivera and Cisternas, 1990; Michael, 1984; Hardebeck & Michael, 2006). I think this should be done before digging deeper into the interpretation of the stress field in and around the ASZ, even though I assume that the results will not dramatically change. The diversity of the focal mechanisms of the individual clusters which is obvious from Fig. 7 and Table 3 fulfills one essential requirement of a successful stress tensor inversion.

We understand your objections about our method and we will include a stress inversion in the revised manuscript.

From my point of view the paper clearly deserves publication in Solid Earth after some minor revisions or comments which I have listed above.

Please also note the supplement to this comment:
https://se.copernicus.org/preprints/se-2020-167/se-2020-167-RC2-supplement.pdf

---

## Author Response (AR1)

**Reply to Referees**

We thank the referees for their comments and suggestions which surely help to clarify our manuscript. In the following we explain how we incorporated their comments into the revised version.

**Answer to Anonymous Referee #1, our answers are written in blue:**

Regarding the region under study, I would recommend to the authors to give more informations on the initial catalog. First, some more details on the quality of the locations, especially at depth (fixed or not), and the magnitudes. Second, it is important to mention if the catalog is discriminated with natural and anthropic events, and how. This is a crucial point.

We added more information about the used catalog in section 3, first paragraph:

As basis for our study we use the earthquake catalog of the LED from 2011 to 2018 for earthquakes within the area close to the ASZ (8.5-9.5° E, 48-48.8° N, Fig. 1b). For these 575 earthquakes we received the bulletin files of the LED (Bulletin-Files des Landeserdbebendienstes B-W, 2018), consisting of hypocenter location, origin time, local magnitude ML and all phase travel time picks with corresponding quality and P-phase polarity. The LED picks from 2011 to 2018 are from 51 LED seismic stations and 14 seismic stations run by other agencies like the state earthquake service of Switzerland (Fig. 1a). Locations are determined with HYPOPLUS, a Hypoinverse variant modified after Oncescu et al. (1996) which allows the usage of a 1.5-D seismic velocity model approach (Stange and Brüstle 2005). Most hypocenter depths could be well determined, but around 9.7% of the depth values were manually fixed. The median uncertainty for longitude, latitude and depth within the catalog are: 0.5 km, 0.6 km, and 2.0 km, respectively. The magnitude ML ranges from 0.0 to 3.4 with average uncertainties of about ±0.2 and the magnitude of completeness is around ML 0.6 (see new Figure S8 in the appendix). The used catalog only contains natural events, as quarry blasts are sorted out and induced events do not occur in the study region.

The catalogue used in this study corresponds to the initial catalog of the LED, complemented by additional pickings. I understand that this would represent another significant work, but I found a bit regrettable to not used the dense temporary network to decrease the completeness magnitude and enlarge the catalogue to get a better spatial distribution of the seismicity with a lower magnitude. Using a simple detection approach and the semi-automatic manual picking would help to improve the dataset.

The detection of small magnitude earthquakes via template matching is current work in progress including location, uncertainty estimation and possible additional fault plane solution determinations. When we did the study presented here, most of our StressTransfer seismic stations only run for a few months in 2018 and we didn't have access to the waveform data of the LED seismic stations yet.

We added a comment section 6:

Thus, we expect another improvement and an increase in detectable events from 2019 onwards due to our additionally installed StressTransfer stations (Fig. 1). Future work will take advantage of the densified seismic station network and focus on small magnitude event detection based on template matching in the area of the ASZ.

In general, the discussion regarding the magnitudes is lacking (also in figure see Figs. 7).

The magnitude values are taken from the LED catalog, these are well determined and cannot be improved within the current uncertainties (±0,2 units). At present we only use magnitude for scaling the size of the fault plane solutions (Fig. 7) and epicenters with magnitude (Fig. 1). As we only took them without changing or complementing anything we don't think a discussion is missing.

In Fig. 1, I am surprised to see so many stations with very few picks.

In Fig. 1 the colors represent the seismic station networks as stated by the legend entries, so permanent (black), StressTransfer (green) or AASN (yellow) and not the number of picks. Maybe you mean Fig. 2? Here, there are many stations with few picks, as Fig. 2 shows the data base used for vp and vs inversion and not the full catalog. The catalog for vp and vs inversion only contains high quality picks, as described in section 4.2 line 174ff. We clarify this in the figure caption of Fig. 2: Seismic stations are indicated as triangles and color-coded with the number of **high quality** picks at a station **used for the vp and vs inversion**.

In Figure 5 as well, the low number of picks (P+S) per event is striking taking into account

the density of the network.

The network shown in Fig 1a is the full network which developed and changed with time. All events in Fig. 5 with yellow colors have 24 and more picks.

I suggest to add the focal mechanisms with the polarities in the Suppl. Mat.

Such a figure is now included in the supplement.

Within the text (section 4.4), I would recommend to modify a bit

around L. 311, because we first think that the authors are doing composite mechanisms.

Because we include SH/P amplitude ratios we no longer use composite fault plane solutions in the revised version.

The authors should mention in the text if the phase amplitudes are also taken

into account in the method they used to determine the nodal planes.

No, they are not included, but we now include SH/P ratios for the revised version of the manuscript. (this was also suggested by reviewer 2)

In section 4.5, all the arguments to chose a strike-slip regime should be included here.

We now include a full stress inversion and changed the text accordingly.

I am not very convinced by the method followed to determined ShMax, which is mainly appropriated for large events. For micro-seismicity, it seems more relevant to consider the stress field in the area under study homogeneous and conduct a inversion of the whole set of

focal mechanisms (e.g., Rivera and Cisternas, 1990; Julien and Cornet, 1989, Michael,

1984, 1984; Hardebeck¡aand Michael, 2006). I am not sure that the spatial variations

of the Shmax direction can be relevant here, as mentioned by the authors. The use

of this large diversity of focal mechanisms probably helps to constrain a stress regime

and stress direction.

We understand your objections regarding the method we used to determine the direction of $S_{Hmax}$, as we mainly have small magnitude events. Originally, we planned to invert for the stress field in future work. However, we include the stress inversion in the revised manuscript as suggested. Accordingly, section 4.5 is reworked completely.

The part of the discussion dealing with the fault plane solution is

unclear. Looking at the Fig 12, it seems that reverse slips occur mainly on the most

dipping planes, which can be surprising, and the authors should explain what his the

main point of this figure.

We changed this part completely, also using the stress inversion result etc.

I found that the last section of the discussion could have been strengthened. After reading it, we do not see what is the contribution of this work to better understand the stress field of this region.

This section is completely changed including now the new stress field inversion.

Figures: Figure 1: This figure should be improved by increasing the size of the maps and by adding the corresponding geological background (transparency on shaded topography) since it is quite developed within the texte (section 2). The epicenters are either marked by circles or gray dots, why? I would recommend as well to change the topographic background by adding an illumination on the slope gradients. Add a rectangle on (a) corresponding to the zone of (b). Add Lake Constance. The magnitude scale is not appropriate since the large events are indicated by stars. More structural and tectonic details on (b), it remains unclear.

The map size is increased. We add a separate map with geological background and an illuminated topography as we think that Fig. 1a) is already very full and will become confusing. All seismic events are shown by gray circles, scaled by magnitude, so dots are events where the circle is very small due to a small magnitude. A rectangle is added in Fig. 1a) framing the area of Fig. 1b). Lake Constance was already included in the figure, but we will increase the font size for better visibility. The magnitude scale is shown for the catalog from 2011 to 2018, which is the catalog we will later use in our area. The large events shown as stars happened all before 2011 and are 'historic' events and are not included in the shown earthquake catalog. We will change the figure caption to make it more clear. We show all known and assumed faults in the area of Fig. 1b) but only mark the relevant ones (HZG) and the tectonic features mentioned in the text (LG, SL).

Figure 2: the limits are confusing. I would recommend to keep the same frames than Fig. 1a. Also, the same colors used in Fig. 1a for the stations are also confusing. Use also a shaded topography.

The limits are the same as Fig. 1a) for the area as well as for the topography colorbar. The meaning of the colors of the triangles is clearly stated by the colorbar and the figure caption.

Indicate the station MSS (the white triangle is not visible in Fig 6).

The size of the white triangle is now increased.

Figure 7: the magnitude scale is required

A magnitude scale is now included.

Figure 8: indicate the area of study and indicate differently your data.

Done.

Figure 10 and 11 should be combined into only one.

We tried to combine both figures but due to their different orientations of the long axis this was not successful. Thus we prefer to keep them separately.

Minor comments:

L. 14: tone down this sentence by removing only, some geophysical imaging and paleo-seismological studies could help as well.

Changed:

New: as there are no visible surface ruptures, which may be connected to the fault zone, we study its characteristics by its seismicity distribution and faulting pattern.

L. 45: major damage: what kind?

Now added:

Although the earthquakes caused major damage to buildings such as fractures in walls and damaged roofs or chimneys, no surface ruptures have been found or described (e.g. Schneider, 1971).

L.47: its geometry. Unclear, if you mean geometry of the seismicity, this is not correct. I would change into to derive the geometry, the segmentation of the faulting pattern.

Changed:

For this reason, the ASZ can only be analyzed by its seismicity to derive the geometry, possible segmentation, and faulting pattern.

L. 49: you mean aprior large earthquakes?

Changed:

L. 51: add 'temporary'

Changed:

Since 2005 the seismic station network of the state earthquake service of Baden-Württemberg (LED) was changed and extended (Stange, 2018) and in summer 2015 the installation of the temporary Alp Array Seismic Network (AASN) started (Hetényi et al., 2018).

L. 58: remove minimum here, this is confusing for the Introduction

Changed:

We calculate a new 1-D seismic velocity model and relocate the events.

L. 59: change geometry of the ASZ into the geometry of the fault pattern at depth in the ASZ

Changed

L. 60: remove permanent

Changed:

L. 84: It is not clear if it is the current or past stress field ?

Changed:

The current regional stress field of southwest Germany is dominated by an average horizontal stress orientation of 150° (e.g. Müller et al., 1992, Reinecker et al., 2010, Heidbach et al., 2016) and was determined from focal mechanism solutions, overcoring, borehole breakouts, and hydraulic fracturing (e.g. Kastrup et al., 2004, Reiter et al., 2015, Heidbach et al., 2016).

L. 94: The directions of the current principal stresses are constant, the amplitudes are not constrained.

We changed the sentence to:

The direction of $S_{Hmax}$ in our research area is quite constant, except of an area directly south of the HZG (Albstadt-Truchtelfingen) and an area within the HZG (Albstadt-Onstmettingen). There the $S_{Hmax}$ direction rotates about 20° counterclockwise into the strike direction of the HZG (130°, Baumann, 1986), which may be caused by a reduced marginal shear resistance.

L. 104: en echelon features compensating the displacements of the ASZ: I am not sure to understand.

With en echelon features we mean graben structures which open in a steplike arrangement at the surface above a partly decoupled strike-slip fault (see Reinecker and Schneider 2002 or Tron and Brun 1991). The tectonic model of Reinecker and Schneider (2002) connects the visible graben structures at the surface (like the HZG) with the ASZ.

Changed text:

L 104: Reinecker and Schneider (2002) propose a tectonic model to relate the graben structures with the ASZ below. They apply the result of Tron and Brun (1991) who showed that the movement of a partly decoupled strike-slip fault in the subsurface can generate graben structures at the surface in a steplike arrangement. In the regional tectonic model the graben structures are the HZG, the Rottenburg Flexure, western Lake Constance faults, and the Fildergraben (Reinecker and Schneider, 2002). The ASZ itself is the strike-slip fault, partly decoupled from the surface by a layer of Middle Triassic evaporites in the overlying sedimentary layers (Reinecker and Schneider, 2002). Stange and Brüstle (2005) consider the bottom of the Mesozoic sediments as mechanical decoupling horizon as no earthquakes occur above 2 km depth.

L.112: moderate displacements : recent, Quaternary?

We now added: ca. 5 Myr according to Reicherter et al. (2008)

L. 115: Add (Fig. 1a) after Lake Constance.

Changed:

Wetzel and Franzke (2003) identified a 5-10 km broad zone of lineations along the 9 °E meridian striking predominantly N-S, NW–SW and ENE-WSE pursuable from Stuttgart in the north to Lake Constance in the south (Fig. 1a).

L. 116: change limit into limited

Changed:

L. 116 to 119: these lines should be reformulated, they are unclear.

Changed to:

Along the 9°E meridian Wetzel and Franzke (2003) identified a 5-10 km broad zone of lineations pursuable from Stuttgart in the north to Lake Constance in the south (Fig. 1a). Those lineations strike predominantly N-S, NW-SW and ENE-WSW. The N-S and ENE-WSW striking faults limited the NW-SE striking graben structures, like the HZG (Reicherter et al., 2008). The NW-Se striking faults are expected to be possibly active at intersections with N-S striking faults due to a reduction in shear resistance accompanied by aseismic creep (Schneider, 1979, Schneider, 1993, Wetzel and Franzke, 2003).

L. 127, 173, 435 + caption_Fig12: remove th and nd to the dates.

These are removed now.

L. 128: add respectively

Changed:

L. 128-129: The average: This is coseismic slip rates. Mention from which data they are deduced. The same for the return period, mention briefly how this is estimated.

Added:

The average seismic dislocation rates along the ASZ are on the order of 0.1 mm/a (Schneider, 1993). The return period of M=5 earthquakes along the ASZ is approximately 1000 years (Schneider, 1980, Reinecker and Schneider, 2002). Both estimates are based on historic earthquake records.

L. 132: change in a depth into at a depth

Changed

L. 134: the term extension is confusing here (lateral extend)

Changed to:

The lateral extend of the fault zone in N-S direction is still under debate:

L. 142: at the beginning of the summer.

Changed:

Additionally, within the AlpArray Project (Hetényi et al., 2018), nine seismic stations were installed starting in summer 2015 within 80 km distance to the ASZ, four of them directly around the ASZ (AlpArray Seismic Network, 2015, Fig. 1b).

L. 158-159: Clarify the error boundaries are checked.

Changed:

Finally, the uncertainty boundaries are checked by eye, and the phase pick is done manually between the two uncertainty boundaries.

L. 180-181: give more explanations.

Now changed in the revised manuscript

L. 206 : a few: :

Changed

L. 222: layers

Changed:

L. 226: no event

Changed

L. 239: the Stress Transfer stations from 2018

Changed

L. 367, 368: writhe the same number of decimals.

Changed:

The determined seismic velocities increase with depth and they are well constrained between 2.50 km and 18.25 km depth (Table 1). The layers between -2.00 km to 2.50 km depth are not very stable due to the non-uniform distribution of rays and sources.

L. 429-431: not necessary. And check consistency with the Fig 12, where

rake varies between -65 and +65_, why?

Done, note we redetermined the fault plane solutions using amplitude ratios.

**Answer to Referee 2, our answers are written in blue:**

1. Focal mechanisms and stress field in SW Germany: When talking about stress field in SW Germany (line 83 – 86) it would be appropriate from my point of view to also reference the paper by Plenefisch & Bonjer (1997) and Bonjer (1997) who inferred the stress field of the Southern Rhine graben area by the inversion of 40 focal mechanisms determined by themselves (Bonjer 1997).

We included the mentioned papers.

See section 2: The regional stress field of southwest Germany is dominated by an average horizontal stress orientation of 150° (e.g. Müller et al., 1992, Plenefisch & Bonjer (1997), Reinecker et al., 2010, Heidbach et al., 2016) and was determined from focal mechanism solutions, overcoring, borehole breakouts and hydraulic fracturing (e.g. Kastrup et al., 2004, Bonjer (1997), Plenefisch & Bonjer (1997), Reiter et al., 2015, Heidbach et al., 2016). It is characterized by NW-SE horizontal compression and NE-SW extension (e.g. Kastrup et al., 2004) and developed during late Miocene (Becker 1993).

2. Presentation of the FOCMEC results: I wonder why the authors do not show any figures with the original output of the FOCMEC program, this means stereographic plots with used polarities and the calculated fault planes that are in accordance with the polarities. This is usual practice and gives the reader an impression about the distribution of the input data and the resulting and suitable pairs of fault planes. I suggest here to show a figure with the results for at least two or three earthquakes with

different quality factors (after Table 3). The solutions for the other events or clusters

respectively could be given either in the supplements or event better - since altogether

these are 'only' 14 solutions - in one comprehensive figure in the paper itself. I think

this is a must. I am aware of the fact that figure S5 shows the uncertainties in strike,

dip and rake. This is a nice figure, but it does not replace the figure I proposed above.

Figures with polarities and fault plane solution fits are now included in the revised manuscript, see supplement.

3. Relative weighting and unity weighting: Please, describe shortly in the text, what is

the difference between these two weighting approaches.

New text in section 4.4:

To find the appropriate solution one can allow different types of errors in FOCMEC. We compare the relative weighting mode and the unity weighting mode of the FOCMEC inversion for all events. This is done to explore if the results differ significantly what could mean that they are questionable (Snoke, 2003). In the unity weighting mode each wrong polarity in the FOCMEC solution counts as an error of one. In the relative weighting mode, polarity errors near a nodal plane count less than polarity errors in the middle of a quadrant. Thus the polarity errors are weighted with respect to their distance to the nodal planes. This means a wrong polarity is weighted by the calculated absolute value of the radiation factor (ranging between 0 and 1). For both modes we searched for solutions allowing increasing errors varying systematically the different possible errors. Those are errors in the P-polarities, errors in the SH-polarities and the total error of wrong SH/P amplitude ratios, as well as the error range in which they are expected to be right. For example, we consider the unity weighting mode and an event with P- and SH-polarities. First, we check if we get a solution for zero errors both. If not we increase the allowed errors for the SH-polarities to one, as the SH-polarities are more unsecure than the P-polarities. If this still doesn't get a solution we check for a wrong P-polarity and no wrong SH-polarity. This procedure is done for unity weighting and relative weighting and stopped if a solution is found. To check for a dependency of the result on a single pick, the next runs for higher errors are also determined.

4. SH polarities: In line 306 the authors state that in some cases they could determine

SH polarities. After all these are 27 events according to Table 3. Since these SH

polarities could be determined I assume that one could also determine amplitude ratios

of SH/P and use them as input in the FOCMEC inversion. The use of amplitude ratios

is scheduled in FOCMEC. It puts further constraints on the solutions and provides

an easy way to determine one single solution by minimizing the differences between

measured and calculated ratios. Why didn't you use such ratios, please comment on

this topic. Perhaps the use of amplitude ratios could help in case of a small number of

polarity observations and could made cluster analysis needless.

We now include the amplitude ratios in the revised version of the manuscript and discuss their usage.

5. Missing a real inversion for the stress tensor: P-, T-and B-axis of a focal mechanism

represent the strain axes. Only in case of a new fracture the strain axes automatically

represent the principal stress axes. However, in case of a preexisting zone of weakness,

which is commonly assumed for small size earthquakes, P-, T- and B-axes do

not usually represent the principal stress axes. It is only the direction of the slip vector

which is constrained by the orientation of the principal stress axes and the relative

stress magnitude. An inversion of an ensemble of slip vectors finally allows the determination

of the stress field and the strike of minimum or maximum horizontal stress

axis respectively. I am not really convinced by the way Mader et al. determine their orientations

of SHmax. I wonder why they have not performed a 'real' stress inversion of

their focal mechanisms. There are several programs to do this (e.g. Gephart & Forsyth,

1984; Rivera and Cisternas, 1990; Michael, 1984; Hardebeck & Michael, 2006). I think

this should be done before digging deeper into the interpretation of the stress field

in and around the ASZ, even though I assume that the results will not dramatically

change. The diversity of the focal mechanisms of the individual clusters which is obvious

from Fig. 7 and Table 3 fulfills one essential requirement of a successful stress

tensor inversion.

We now include a stress inversion in the revised manuscript and discuss its usage and implications.

---

## Author Response (AR2)

Dear Christian Sue,

we just changed some formatting in the reference list, changed the reference Mader and Ritter, in print to Mader and Ritter (2021) and included a missing reference for the topography of Figure 1c.

Yours sincerely

S. Mader, J.R.R. Ritter and K. Reicherter